Methods

# pyRBDome: a comprehensive computational platform for enhancing RNA-binding proteome data

Liang-Cui Chu[1,2,*] , Niki Christopoulou[1,2,*] , Hugh McCaughan[1,2,*], Sophie Winterbourne[2] , Davide Cazzola[1] , Shichao Wang[1,2], Ulad Litvin[1,3], Salomé Brunon[1,4] , Patrick JB Harker[1,5] , Iain McNae[2], Sander Granneman[1,2]

High-throughput proteomics approaches have revolutionised the identification of RNA-binding proteins (RBPome) and RNA-binding sequences (RBDome) across organisms. Yet, the extent of noise, including false positives, associated with these methodologies, is difficult to quantify as experimental approaches for validating the results are generally low throughput. To address this, we introduce pyRBDome, a pipeline for enhancing RNA-binding proteome data in silico. It aligns the experimental results with RNA-binding site (RBS) predictions from distinct machine-learning tools and integrates high-resolution structural data when available. Its statistical evaluation of RBDome data enables quick identification of likely genuine RNA-binders in experimental datasets. Furthermore, by leveraging the pyRBDome results, we have enhanced the sensitivity and specificity of RBS detection through training new ensemble machine-learning models. pyRBDome analysis of a human RBDome dataset, compared with known structural data, revealed that although UV–cross-linked amino acids were more likely to contain predicted RBSs, they infrequently bind RNA in high-resolution structures. This discrepancy underscores the limitations of structural data as benchmarks, positioning pyRBDome as a valuable alternative for increasing confidence in RBDome datasets.

## Introduction

RNA-binding proteins (RBPs) play diverse and crucial roles in gene expression by influencing the structure, function, and stability of RNA, both co- and post-transcriptionally (Glisovic et al, 2008; Holmqvist & Vogel, 2018). RBPs have been associated with many human diseases, including neurological disorders, muscular atrophies, and cancer (Castello et al, 2013). In bacteria, RBPs make key contributions to rapid adaptation to challenging environments, and in pathogens, they control virulence and the capacity for host infections (Holmqvist & Vogel, 2018; Christopoulou & Granneman, 2022). Because of their key functions, considerable efforts are being made to identify RBPs in diverse organisms and to characterise these proteins functionally and structurally. This has inspired the development of several high-throughput methods that capture all proteins interacting with RNA (RBPome). These methods usually involve UV or chemical treatment of cells to create covalent bonds between proteins and RNA substrates. This is followed by enrichment of the cross-linked RNA–protein complexes and identification of proteins by quantitative mass spectrometry (MS) (reviewed in Esteban-Serna et al [2023]). Common approaches for enriching RNA–protein complexes include using oligo(dT) beads to capture proteins cross-linked to polyadenylated RNAs (Baltz et al, 2012; Castello et al, 2012, 2016; Stenum et al, 2023), silica beads that capture all RNAs and cross-linked proteins (Beckmann et al, 2015; Asencio et al, 2018; Shchepachev et al, 2019; Trendel et al, 2019; Bae et al, 2020; Chu et al, 2022), or organic–aqueous phase separation methods that rely on the fact that cross-linked RNAs alter the physiochemical properties of proteins (Queiroz et al, 2019; Trendel et al, 2019; Urdaneta et al, 2019; Smith et al, 2020). To identify the cross-linked proteins, purified complexes are treated with ribonucleases and analysed by MS.

These ground-breaking studies have uncovered a plethora of novel RBPs in diverse organisms, many of which contain domains that have never been associated with RNA-binding before. Although having a comprehensive list of all RBPs in your favourite organism is tremendously valuable, the next most informative piece of information would be the location of the RNA-binding domains (RBDs) within these proteins (RBDome), as this would allow mechanistic insights into RNA recognition and the design of mutations to dissect the physiological significance of RNA-binding. Although protocols for the global identification of putative RBPs have been optimised for diverse organisms, identifying the amino acid sequences UV–cross-linked to RNA (and therefore likely directly bind RNA in vivo) in RBPome data is both experimentally and computationally challenging. To identify amino acid–RNA adducts,

[1]Centre for Engineering Biology, University of Edinburgh, Edinburgh, UK  [2]Institute of Quantitative Biology, Biochemistry and Biotechnology, University of Edinburgh, Edinburgh, UK  [3]MRC-University of Glasgow Centre for Virus Research, Glasgow, UK  [4]Institut de Biologie de l'Ecole Normale Supérieure (IBENS), Paris, France  [5]Cancer Research UK Cancer Biomarker Centre, University of Manchester, Manchester, UK

Correspondence: Sander.Granneman@ed.ac.uk
*Liang-Cui Chu, Niki Christopoulou, and Hugh McCaughan contributed equally to this work

the cross-linked RNA is chemically or enzymatically digested to make detection of the cross-linking site by MS feasible. However, this digestion is often incomplete, and the heterogeneity in the length and sequence of nucleotide adducts generates variable mass shifts. This dramatically increases the MS/MS search space, making detection of the cross-linking sites using conventional MS data analysis programs unfeasible. To overcome this problem, several experimental computational MS workflows have been developed that either directly detect peptide–RNA conjugates (Schmidt et al, 2012; Kramer et al, 2014; Kong et al, 2017; Trendel et al, 2019; Yu et al, 2020; Götze et al, 2021; Knörlein et al, 2022) or identify putative RNA-binding sites (RBSs) by relying on the fact that sequences neighbouring the cross-linked peptides *can* be identified by conventional MS (RBDmap [Castello et al, 2016]), allowing extrapolation of sequences most likely cross-linked to RNA. Recent RBDome methods (RBS-ID and pRBS-ID) use hydrofluoride to chemically digest RNAs cross-linked to peptides to a single nucleotide (Bae et al, 2020, 2021). This greatly reduces the computational workload, increasing the sensitivity of cross-linking site detection at a single amino acid resolution (Bae et al, 2020, 2021).

Although RBDome and RBPome methods have generated a wealth of valuable data, each has its own caveats and noise levels. Thus, there is a possibility of recovering many false-positive hits (Nesvizhskii et al, 2006; Bogdanow et al, 2016; Bae et al, 2020). For example, although RBDome methods promise a single amino acid resolution of binding site identification, there is a degree of uncertainty when it comes to mapping the cross-linked amino acid (Edwards, 2013; Kim & Pevzner, 2014; Bae et al, 2020). Moreover, a recent study has shown that UV–cross-linked amino acids detected by these methods can also be indirectly cross-linked to RNA (Knörlein et al, 2022). Evidently, experimental validation of the findings is critical; however, the available methodologies are generally low throughput, making it challenging to quantify what fraction of RBDome data are biologically meaningful. An alternative approach would be to enhance the reliability of the experimental results using computational approaches. For example, one could calculate what fraction of cross-linked amino acids in RBDome data are in known RBDs (Queiroz et al, 2019; Bae et al, 2020, 2021) or interact with RNA in available crystal structures (Knörlein et al, 2022). To conduct a meaningful statistical analysis, however, a ground truth dataset is required that (ideally) consists of a large collection of high-resolution structures of protein–RNA complexes. However, such datasets are not readily available, especially for model organisms for which few protein–RNA complexes have been structurally characterised. This includes one of our favourite model organisms: *Staphylococcus aureus*. Furthermore, although extremely informative, ground truth datasets are not exhaustive, as they generally only contain relatively stable interactions that can be structurally characterised and interactions with single RNA substrates.

As an alternative, but also complementary, approach for assessing and enhancing the quality of experimental RBPome and RBDome data, we developed a Python computational pipeline (pyRBDome). This pipeline compares results from these high-throughput analyses against a large database of predicted RBSs. The pipeline generates this database for proteins of interest using a wide variety of different prediction tools that use distinct approaches for predicting RBSs. Subsequently, the pipeline aggregates the results and putative RBSs

are superimposed on (model) structures and other human-readable formats. When provided with RBPome data, the pipeline enables users to extract the most likely RNA-binders and identify amino acids most likely to bind RNA. When also provided with a list of cross-linked peptides (RBDmap, RBDome data) and amino acids (RBDome data), pyRBDome identifies the most common peptide motifs associated with RNA-binding and determines whether the data are significantly enriched for predicted RBSs by calculating 3D distances between experimental and predicted RBSs. By displaying Pfam domains (Mistry et al, 2021) identified in 3D structures, the user can easily determine the domains involved in the interactions. By clustering the cross-linking sites/peptides in domain structures, pyRBDome can identify interfaces within domains involved in RNA-binding. In conclusion, pyRBDome can reveal important mechanistic insights into RNA recognition, greatly facilitating further experimental validation of RNA-binding.

A second and equally important motivation for developing this pipeline was to make the analysis of RBP/RBDome datasets more accessible to groups that do not routinely perform such experiments or wish to analyse existing datasets. Moreover, because the pyRBDome code was written as Python classes with associated test Jupyter notebooks, these can also be readily incorporated into new software tools.

Here, we demonstrate how pyRBDome can effectively identify putative RNA-binding sequences in human and bacterial proteins and enhance RBDome datasets computationally. Moreover, using machine learning (ML), we show that combining prediction results from distinct computational tools employed in pyRBDome can substantially enhance the sensitivity and specificity of computational prediction of RBSs in RBPs. We provide a detailed comparison of the experimental and pyRBDome data with known structural data, obtained from human protein–RNA complexes and *Streptococcus pyogenes* Cas9 ribonucleoprotein complexes. These analyses revealed that UV cross-linking sites in proteins often correlate with proximity to RNA in structurally characterised protein–RNA complexes, although not necessarily with direct RNA interaction.

## Results

### The pyRBDome pipeline

The main goal of this project was to develop a pipeline that would enable us to evaluate and enhance the quality of RBPome and RBDome datasets. The pyRBDome pipeline is written in Python, and the various analysis steps are provided in a series of Jupyter notebooks to facilitate the process of following, controlling, and adjusting the analysis steps. The pipeline consists of two parts: pyRBDome-Core and pyRBDome-Notebooks. The former contains the Python classes and functions that are required for running the pyRBDome-Notebooks code. Each class in pyRBDome-Core has associated test Jupyter notebooks, making it easy to learn how to run the code. This should facilitate incorporation of the code into new bioinformatics tools. All the notebooks can be run either in Jupyter or in the terminal using papermill (https://papermill.readthedocs.io/en/latest/). A schematic representation of the entire pipeline is shown in Figs 1 and S1. A minimum requirement for running the pipeline is an Excel file with a

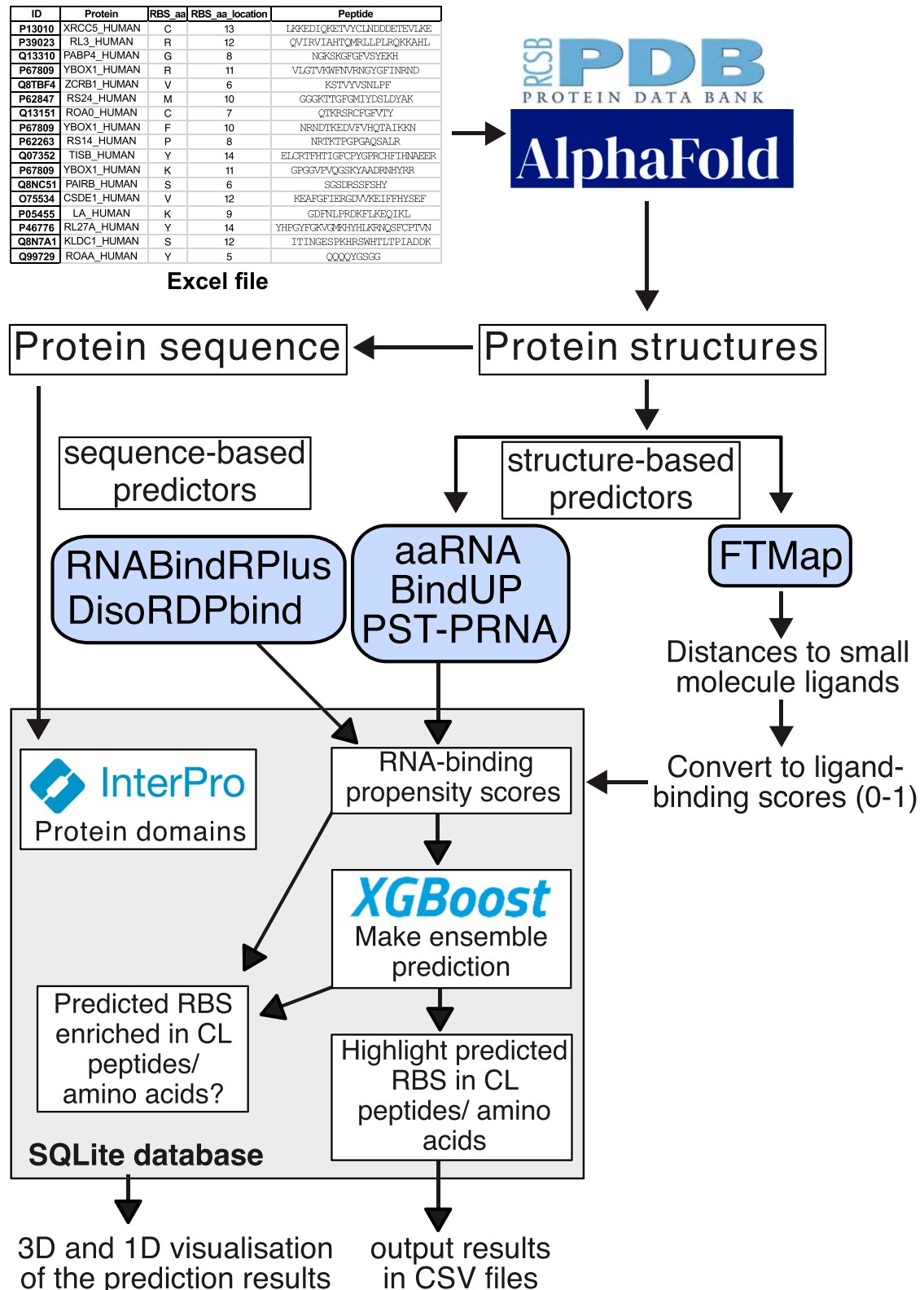

**RBPome or RBDome data**

| ID | Protein | RBS_aa | RBS_aa_location | Peptide |
|---|---|---|---|---|
| P13010 | XRCC5_HUMAN | C | 13 | LKKEDIQKETVYCLNDDDETEVLKE |
| P39023 | RL3_HUMAN | R | 12 | QVIRVIAHTQMRLLPLRQKKAHL |
| Q13310 | PABP4_HUMAN | G | 8 | NGKSKGFGFVSYEKH |
| P67809 | YBOX1_HUMAN | R | 11 | VLGTVKWFNVRNGYGFINRND |
| Q8TBF4 | ZCRB1_HUMAN | V | 6 | KSTVYVSNLPF |
| P62847 | RS24_HUMAN | M | 10 | GGGKTTGFGMIYDSLDYAK |
| Q13151 | ROA0_HUMAN | C | 7 | QTKRSRCFGFVTY |
| P67809 | YBOX1_HUMAN | F | 10 | NRNDTKEDVFVHQTAIKKN |
| P62263 | RS14_HUMAN | P | 8 | NRTKTPGPGAQSALR |
| Q07352 | TISB_HUMAN | Y | 14 | ELCRTFHTIGFCPYGPRCHFIHNAEER |
| P67809 | YBOX1_HUMAN | K | 11 | GPGGVPVQGSKYAADRNHYRR |
| Q8NC51 | PAIRB_HUMAN | S | 6 | SGSDRSSFSHY |
| O75534 | CSDE1_HUMAN | V | 12 | KEAFGFIERGDVVKEIFFHYSEF |
| P05455 | LA_HUMAN | K | 9 | GDFNLPRDKFLKEQIKL |
| P46776 | RL27A_HUMAN | Y | 14 | YHFGYFGKVGMKHYHLKRNQSFCPTVN |
| Q8N7A1 | KLDC1_HUMAN | S | 12 | ITINGESPKHRSWHTLTPIADDK |
| Q99729 | ROAA_HUMAN | Y | 5 | QQQQYGSGG |

**Excel file**

**Protein sequence** ← **Protein structures**

sequence-based predictors

structure-based predictors

**RNABindRPlus DisoRDPbind**

**aaRNA BindUP PST-PRNA**

**FTMap**

Distances to small molecule ligands

Convert to ligand-binding scores (0-1)

**InterPro**
Protein domains

RNA-binding propensity scores

**XGBoost**
Make ensemble prediction

Predicted RBS enriched in CL peptides/ amino acids?

Highlight predicted RBS in CL peptides/ amino acids

**SQLite database**

3D and 1D visualisation of the prediction results

output results in CSV files

list of UniProt IDs for their proteins of interest. The pipeline will then enable users to identify putative RBSs within these proteins. If a list of putative RNA-binding peptides or amino acids for these UniProt IDs was also provided, such as data from RBDmap (Castello et al, 2016) or RBS-ID (Bae et al, 2020, 2021), the pipeline will enable the user to identify which among the provided sequences/amino acids contains predicted RBSs, enabling effective selection of sequences that are most likely to bind RNA. An example of such a CSV input file is provided in Table S1. To facilitate these analyses, pyRBDome relies on multiple distinct RBS prediction tools. Considering the large size of RBS-ID and RBDmap data, and therefore the need to process a substantial number of proteins within a reasonable timeframe, the selection of these tools (see Table 1) was based not only on their performance, but also on their runtime, and the ability to submit many proteins to web servers (also see the Discussion section).

RBS predictions are generally based on a wide range of features, such as amino acid sequence, structural data, and physicochemical properties of the studied proteins. Three of the computational programs used were specifically designed to identify potential RBSs using protein structure (aaRNA and PST-PRNA [Li et al, 2014; Li & Liu, 2022]) and/or purely sequence information (RNABindRPlus [Walia et al, 2014]). However, a potential limitation of using these programs is that they were trained on data from known RBPs, which might make them less effective in identifying RBS in unconventional RBPs. Therefore, we also analysed our data using BindUP, which predicts RBSs based on the electrostatic features on the protein surface and can more reliably detect non-canonical RBPs (Paz et al, 2016). RBSs can sometimes overlap with small molecule binding sites of enzymes, such as in the case of GAPDH, aconitase (Walden et al, 2006), and thymidine synthase (Chu et al, 1991). Hence, we used FTMap (Brenke et al, 2009) to find putative small molecule binding sites in structures. FTMap identifies possible ligand-binding pockets by globally docking a series of small organic probes onto the input structures to identify protein regions that represent binding hotspots. Incorporating FTMap data also offers the additional benefit of enabling the selection of RBPs with a higher likelihood of being druggable. In addition, many RBPs contain flexible and/or disordered domains, which are common in eukaryotic species. Therefore, we also included DisoRDPbind (Peng & Kurgan, 2015), which predicts RBSs in intrinsically disordered regions.

Consequently, pyRBDome integrates several independent yet complementary computational methodologies (see Table 1 for an overview of the tools) to compare against biochemically derived RNA-interacting protein sequences. Although each approach has its own degree of uncertainty, our rationale lies in the consistency across these methods to identify amino acids more likely to be bona fide RBSs.

Several of the aforementioned tools rely on structural data to make their predictions (Fig 1). If available, the pipeline automatically downloads these structures from rcsb.org. In cases where such information is unavailable, pyRBDome retrieves structural estimates generated by AlphaFold2 (Jumper et al, 2021). This facilitates the analysis of RBPome and RBDome data from less well-characterised model organisms. By default, the program will download AlphaFold2 models as these contain the complete protein sequence, including disordered regions (even if they are not properly folded).

To compare the experimental data with the predictions, for each peptide sequence provided (Fig 1), the pipeline calculates the minimal distance (in Å) to RBSs predicted by the individual tools. It stores its progress, such as whether files have been downloaded from web servers or specific tasks have been completed, as well as the analysis results in a SQLite database (Fig 1). The final results can subsequently be exported to CSV files where for each cross-linked peptide (Table S2) or amino acid (Table S3) provided, the pipeline reports where in the PDB file the peptide was mapped to and how frequently a predicted RBS was detected. Manual inspection of the data in PyMOL (Schrödinger, 2024) revealed that cross-linked peptides and amino acids were often found near known RBSs. Therefore, we consider cross-linked sequences (peptides or amino acids) that are in proximity of predicted sites (within hydrogen-bonding distance (4.2 Å) as a starting point) as promising hits. All the results of the analyses are then summarised in a large table, containing the prediction results from each tool for each amino acid in the analysed proteins (Table S4). In addition, if structural information is available for any proteins in complex with RNA, the user can also instruct the pipeline to measure the distance to RNA molecules in known structures and whether the amino acid is known to bind RNA (Table S5). Finally, using InterProScan (Quevillon et al, 2005), locations of domains within the protein sequences are determined, making it possible to identify domains involved in RNA-binding. The tables generated by the pipeline make it straightforward to statistically identify sequences obtained from RBDome experiments that are more likely to be bona fide RNA-binders.

## Building ground truth datasets to compare experimental and prediction data with structurally analysed protein–RNA complexes

To showcase the feasibility of pyRBDome, we applied the pipeline to the recent RBS-ID RBDome dataset (Bae et al, 2020). This dataset was chosen because at the start of this project, it was the richest

---

**Figure 1. Schematic representation of the pyRBDome pipeline.**
To run the pipeline, the user needs to provide a table, with the same column names as shown in the table, that includes the UniProt identifier (ID), the name of the protein, cross-linked amino acids (if available), cross-linked peptides (if available), and the location of the cross-linked amino acid in the corresponding cross-linked peptide sequence. This table is then used to download structural information from rcsb.org or from AlphaFold2. The protein structures (in PDB files) are then submitted to various web servers that predict ligand-binding sites on the protein. The protein sequences are extracted from the PDB files and submitted to prediction algorithms that use sequence-based information to predict RNA-binding sites (RBSs). Domain information is also extracted from the protein sequences, using the InterProScan tool. Once all the predictions have been completed, the pipeline gathers all the data by collating the results in a SQLite database (see Fig S6 for an example of such a table). This is then fed to our XGBoost ensemble model to predict RBS, with the aim of further enhancing the detection of RBSs. The resulting data are then highlighted within the provided cross-linked peptide sequence. Moreover, statistical analyses are performed to determine whether cross-linked peptides/amino acids are enriched for predicted RBSs.

# Life Science Alliance

**Table 1.   Ligand-binding site prediction tools employed by pyRBDome, including a brief description of their functionality.**

| Tool | Purpose | Reference |
|---|---|---|
| RNABindRPlus | Predicts RNA-binding amino acids within proteins using protein sequences by combining output from support vector machines (SVMs) with outputs from sequence homology–based methods | Walia et al (2014) |
| BindUP | Predicts DNA- and RNA-binding regions using structural information. BindUP predicts RNA-binding regions solely based on the presence of electrostatic patches on the surface of the protein. It does not rely on any sequence or structural conservation/homology | Paz et al (2016) |
| PST-PRNA | A deep learning approach that predicts RNA-binding sites based on protein surface topography (PST). It relies on structural information for the protein of interest | Li and Liu (2022) |
| aaRNA | A machine-learning method that predicts RNA-binding sites using both sequence and structural information. Predictions are based on sequence conservation, surface deformations, and relative solvent accessibility. Note that this server is no longer online and will therefore be removed from future versions of pyRBDome. | Li et al (2014) |
| DisoRDPbind | A deep learning approach that predicts RNA-, DNA-, and protein-binding residues in intrinsically disordered regions in proteins. It uses protein sequence information to extract physiochemical properties of the protein, putative secondary structure, and disorder to make its predictions. | Peng and Kurgan (2015) |
| FTMap | Identifies ligand-binding hotspots in proteins for drug discovery purposes. | Brenke et al (2009) |

cross-linking dataset available: it includes data for almost 600 human RBPs, including cross-linked amino acids that could represent RBSs. In addition, this dataset contains a rich RBDome dataset for the *S. pyogenes* Cas9 protein (spCas9) in complex with RNA, which we use throughout this study to showcase our pyRBDome analysis results.

To facilitate the comparison of experimental data with predictions, pyRBDome requires peptide sequences that are at least four amino acids long as it needs to locate these sequences in 3D (model) structures. However, because the published RBS-ID data only provided the locations of cross-linked amino acids, we artificially extended these sequences on both ends with varying lengths (up to 27 amino acids; arbitrary number) to generate a dataset that we refer to as the "cross-linked peptide" dataset. The results of the pyRBDome analyses of this dataset are organised in tabular form in Table S4.

Another reason for choosing the RBS-ID dataset was that high-resolution protein–RNA structures were available for 155 of the ~600 human proteins it includes. Similarly, structural data for spCas9-RNA complexes were also available (Anders et al, 2014). Consequently, we could compare the RBS-ID results with both RBS predictions collated by the pyRBDome pipeline and known protein–RNA interactions (ground truth dataset). To establish such human ground truth datasets, we downloaded hundreds of PDB files containing human protein–RNA complexes from rcsb.org. This yielded 371 protein–RNA structures (including the 155 present in the RBS-ID data) that met our criteria for downstream analyses (see the Materials and Methods section for details). Using these structures, we generated two distinct

ground truth datasets. As an illustrative example, the analyses for a spCas9-RNA complex (Anders et al, 2014) are shown in Fig 2. Firstly, we used protein–ligand interaction profiler (PLIP [Adasme et al, 2021]) to identify amino acids directly interacting with RNA in these structures. This ground truth dataset is referred to as GT-PLIP (Fig 2A). The PLIP software package also enabled us to identify specific types of protein–RNA interactions, such as hydrogen-bonding (Fig 2B), π-stacking, hydrophobic, and salt bridge interactions (Fig 2C).

When we analysed the entire RBS-ID dataset with PLIP, because of limitations in resolution, not all structures generated results, yielding a relatively small dataset comprising 192 proteins. To address this (potential) limitation, we established a second ground truth dataset, categorising amino acids that are within hydrogen-bonding distance (≤4.2 Å) of RNA as RNA-binding (0 for non-interacting and 1 for interacting amino acids). We refer to this ground truth dataset as GT-Distance (spCas9 example shown in Fig 2D). This generated a richer and larger dataset (n = 347), with ~10% of the amino acids assigned as RNA-interacting. To capture all experimentally determined protein–RNA interactions for each protein, PLIP and distance-based detection of RBSs were performed using all available protein–RNA structures associated with individual UniProt IDs. Subsequently, the analysis results from multiple PDB files for a protein were merged into a single PDB file that stored for each amino acid the minimal distance to RNA and how frequently binding to RNA was detected.

Having these ground truth datasets also allowed us to benchmark the different prediction tools employed in pyRBDome and to directly compare their performances. To calculate key performance

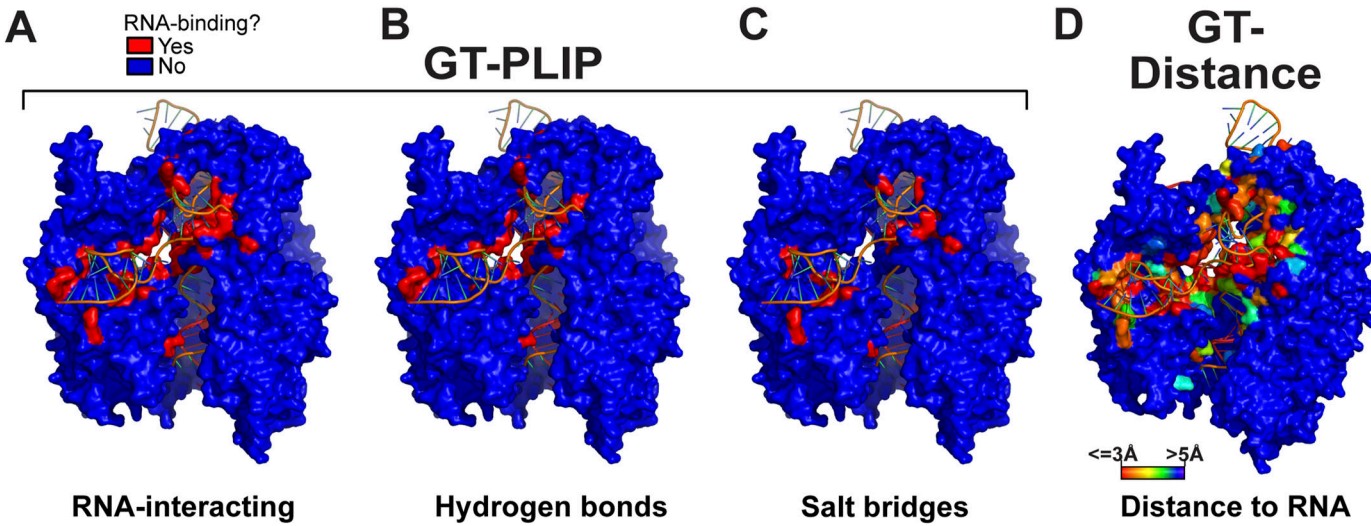

**Figure 2. Ground truth analysis results for the *Streptococcus pyogenes* Cas9 protein.**
Shown is a surface representation of the structure of the spCas9 protein in complex with cocrystallised RNA and DNA (orange colour), obtained from rcsb.org (PDB 4un3 [Anders et al, 2014]). **(A, B, C)** Results of the protein–ligand interaction profiler analysis (Adasme et al, 2021) on the spCas9-RNA interactions. Shown are the results for all the RNA-binding residues (A), the amino acids that form hydrogen bonds with RNA (B), and those that form salt bridges (C). Blue amino acids indicate those that do not bind RNA directly. Red amino acids indicate those that do. **(D)** Highlighting amino acids that are in proximity to RNA in the spCas9-RNA complex. To generate the GT-Distance ground truth dataset, we considered amino acids that are within 4.2 Å of RNA in the available structures as RNA-binding. Those amino acids closest to RNA are highlighted in red, whereas those >4.2 Å from RNA are highlighted in blue.

metrics, we used our ground truth datasets and recommended probability/scoring thresholds for identifying an amino acid as RNA-binding (Brenke et al, 2009; Li et al, 2014; Walia et al, 2014; Peng & Kurgan, 2015; Paz et al, 2016; Li & Liu, 2022). The results of these analyses for each predictor are provided in Fig S2. This revealed that PST-PRNA is the best performing tool on both the GT-PLIP (Fig S2A) and GT-Distance (Fig S2B) datasets, achieving the highest accuracy and precision. RNABindRPlus, the sequence-based prediction algorithm, followed closely in second place. Notably, the performance of aaRNA on our GT-Distance dataset was comparable to its performance on a smaller ground truth dataset consisting of 67 RBPs (RB67 [Li et al, 2014]).

To simplify and automate the generation of ground truth datasets, we have included scripts in pyRBDome-Core that contain code needed for automated downloading of protein (FindUni-ProtPDBStructures.py) and protein–RNA complexes (FindUni-ProtRNPStructures.py) associated with specific UniProt IDs from rcsb.org, as well as code to calculate the distances of each amino acid to RNA (ProteinNAdistanceAnalyses.py). We also wrote code to automate the PLIP analysis and the processing of the analysis results (https://git.ecdf.ed.ac.uk/sgrannem/pyDRBPNA). All the results generated by our ground truth analysis code are summarised in Table S5.

### Visualisation of the pyRBDome analysis results

To aid and simplify the interpretation of the results, the pipeline generates PyMOL session files after the completion of the analyses (Figs 1 and S1). These files facilitate the visual representation of prediction outcomes on the structural data in PDB files and allow viewing of all results simultaneously. The RNA-binding propensities

provided by each prediction algorithm are included in b-factor columns of PDB files. By colouring the atoms according to their b-factor values in PyMOL, users can then readily identify amino acid residues with the highest RNA-binding propensities. These results also make it possible to identify putative RNA-binding interfaces on proteins of interest. In addition, the pipeline can produce PDF files containing the aligned prediction results within the protein sequence, including any domains detected by InterProScan, experimental RBDome data, and any available structural data. To generate the PDF files, the pipeline again uses recommended RNA-binding propensity thresholds (Brenke et al, 2009; Li et al, 2014; Walia et al, 2014; Peng & Kurgan, 2015; Paz et al, 2016; Li & Liu, 2022) to call whether an amino acid binds RNA.

To illustrate the outputs from the pyRBDome pipeline, we superimposed the results for the spCas9 protein on the spCas9-RNA complex structure (Anders et al, 2014). The pipeline highlights the individual protein domains in a PDB file (Fig 3A) and indicates where cross-linked peptides (Fig 3B) and amino acids (Fig 3C) are in the structure. Fig 3D–F shows the PST-PRNA, BindUP, and RNABindRPlus results for the spCas9 protein. An overview of all the spCas9 prediction results, as well as the available RBDome and structural data, in the protein sequence can be found in Fig S3. This figure also highlights amino acids in a spCas9-RNA complex (Anders et al, 2014) that bind RNA or are within hydrogen-bonding distance of RNA.

### Identification of domains and amino acids preferentially cross-linking to RNA

Previous studies have shown a bias in the type of amino acids that cross-link to RNA (Shchepachev et al, 2019; Bae et al, 2020). Therefore, one way to assess the quality of RBDome data is to

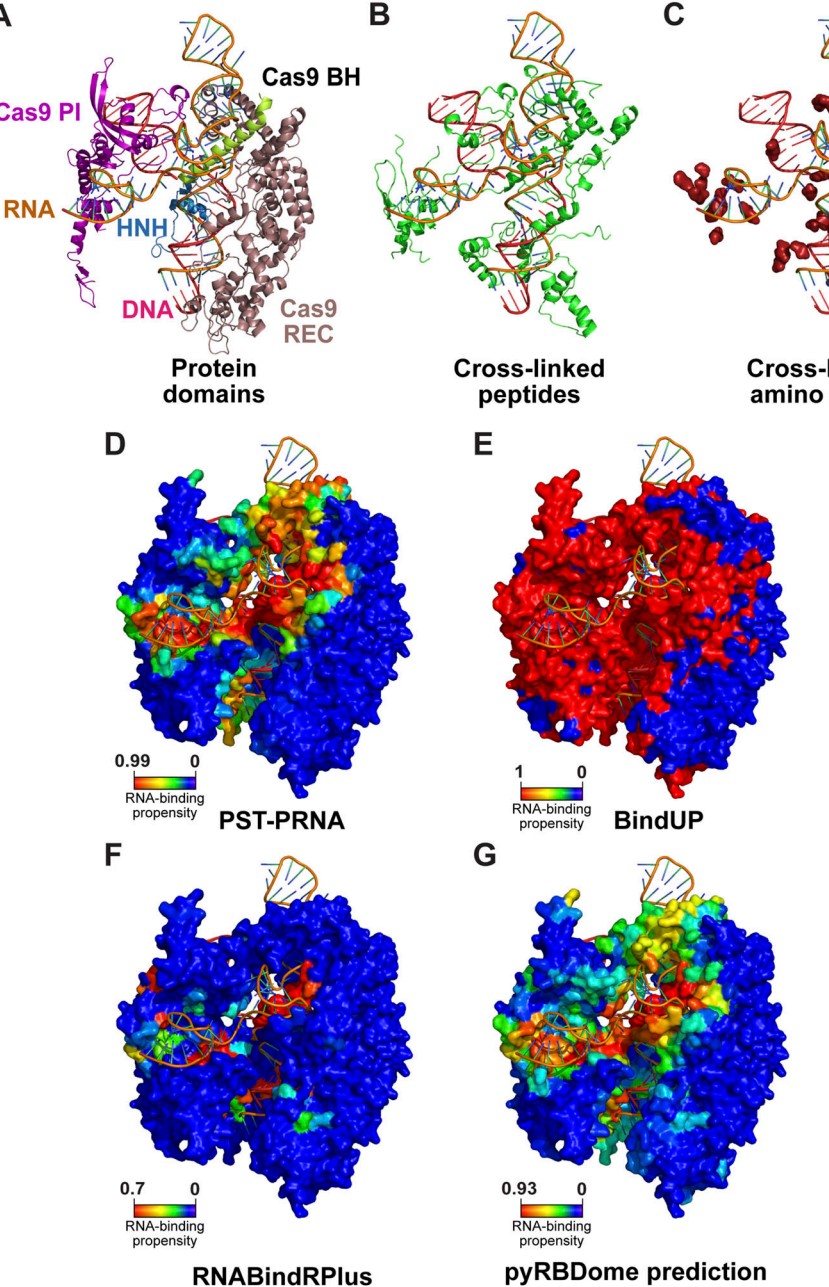

**Figure 3. pyRBDome analysis results for the spCas9 protein.**
**(A)** Shown is the structure of the RNA and DNA molecules within the structure, as well as the individual spCas9 protein domains detected by InterProScan (Jones et al, 2014). **(B)** Same as in (A) but now with the location of cross-linked peptides within the structure. **(C)** Same as in (B) but now with the location of the cross-linked amino acids (shown as surface, red colour). **(D, E, F, G)**. Examples of prediction results from various tools employed by pyRBDome. Shown is a surface representation of the spCas9 protein, with nucleic acids shown in orange. Accompanying colour bars represent the RNA-binding propensities, correlating specific colours with their respective values.

determine whether similar amino acid cross-linking preference can be detected in the data. Code to perform these analyses is provided in pyRBDome-Notebooks. Consistent with previous analyses (Bae et al, 2020), pyRBDome identified cysteines and the aromatic amino acids tyrosine, tryptophan, and phenylalanine as the most cross-linked amino acids in the RBS-ID data (Fig S4A). In addition, the pipeline performs a second enrichment analysis by grouping the amino acids into bins based on their physicochemical properties (Fig S4B). This identified sulphur-containing and aromatic amino acids as preferentially cross-linked. pyRBDome also enables the user to determine whether sequences from specific domains were preferentially cross-linked. Using the InterProScan package (Jones et al, 2014; Blum et al, 2021), pyRBDome searches for domains within the proteins identified in the experimental data and it then counts how frequently cross-linked peptides and amino acids were mapped to these domains. Consistent with the previous work (Bae et al, 2020), the canonical RNA recognition motif (RRM) and hnRNP K homology (KH) RBDs were the most enriched domains in the cross-linking data, followed by zinc finger (ZnF: $C_2H_2$, CCCH, and CCHC), WD40 repeats, and helicase/DEAD domains (Fig S4C).

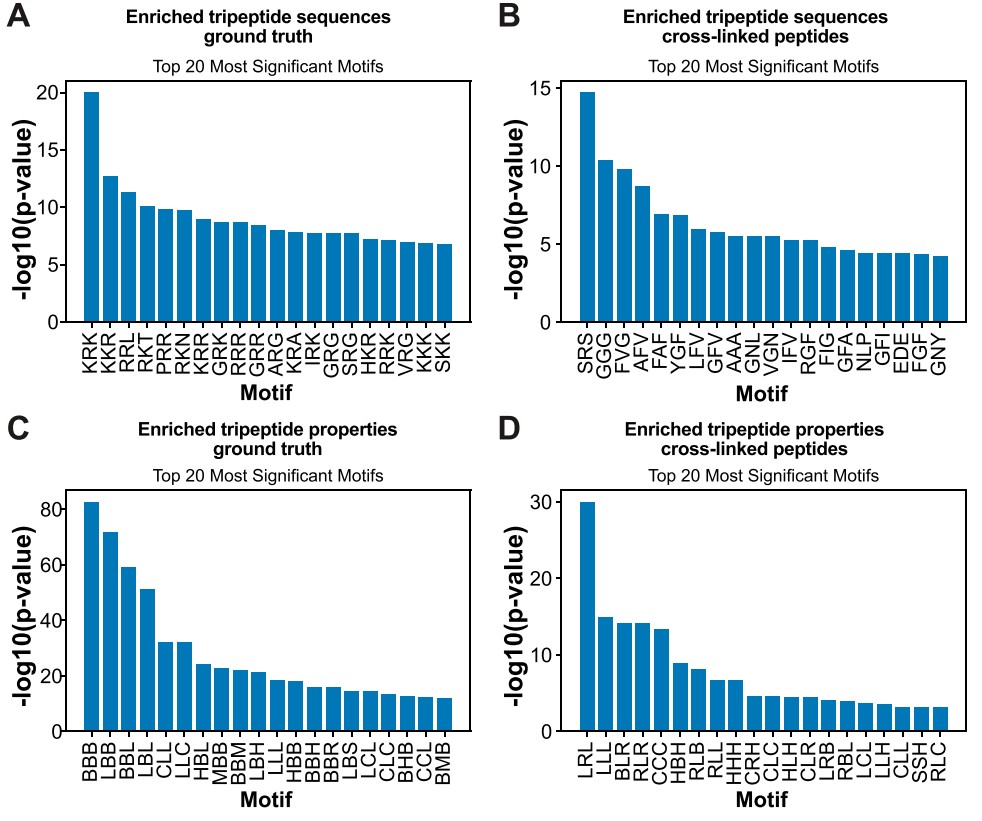

**Figure 4. Cross-linked peptides are enriched for tripeptides containing aromatic and positively charged amino acids flanked by aliphatic residues.**
**(A)** Tripeptide motifs detected in RNA-binding regions (amino acids within 4.2 Å from RNA) from known RNA-binding proteins. **(B)** Tripeptide motifs enriched in the RBS-ID cross-linked peptides. **(A, C)** Enriched chemical properties of tripeptide sequences detected in the ground truth data described in (A). **(D)** as in (B) but now showing the chemical properties. Categories: L, aliphatic; R, aromatic; C, acidic; B, basic; H, hydroxylic; S, sulphur-containing; M, amidic. *P*-values were calculated using the Fisher exact test and corrected for multiple testing using the Benjamini–Hochberg procedure.

## UV irradiation favours cross-linking RNA to positively charged and aromatic amino acids flanked by aliphatic residues

The likelihood of an RNA–protein interaction at a specific site is significantly influenced not only by the chemical properties of amino acids but also by its neighbours, owing to favourable protein folding or surface electrostatic forces. Recent studies have demonstrated that RBPs are enriched for tripeptide motifs consisting of positively charged, negatively charged, and aliphatic amino acids, and these triplets are conserved across evolution (Beckmann et al, 2015; Bressin et al, 2019). In three organisms that were analysed (*Homo sapiens*, *Escherichia coli*, and *Salmonella typhimurium*), tripeptides with a combination of arginines, lysines, and glycines were strong predictors of RBPs. The pyRBDome pipeline can perform tripeptide motif analyses of RBDome data, enabling users to identify motifs most likely to contribute to RNA-binding in their model organism. pyRBDome searches for tripeptide motifs enriched in the cross-linked peptides relative to randomly selected peptides from the same protein sequence (Fig 4A). To enhance these analyses, pyRBDome also performs the same motif analyses based on the biochemical properties of the amino acids in the tripeptide motifs (Fig 4C). Strikingly, the results show that although amino acids with positively charged residues are highly enriched in the human ground truth data (Fig 4A and C), tripeptides containing combinations of aromatic (i.e., Y and F) and aliphatic (i.e., G, V, and A) are very highly enriched in the cross-linked peptides (Fig 4B and D). This is consistent with the strong

bias towards UV cross-linking to specific amino acids, such as aromatic amino acids, to RNA.

## pyRBDome reveals insights into domain RNA-binding interfaces

In addition to providing information about enriched domains in RBDome data, the pipeline can also identify RNA-binding interfaces within individual domains. UV cross-linking is inefficient and stochastic, so within individual protein domains, only a few of all possible RNA-binding interactions will be detected, providing limited mechanistic insights into domain–RNA interactions. However, it is reasonable to assume that these domains within different proteins will have defined modes of RNA recognition. Therefore, if peptides/amino acids reported in RBDome data indeed represent genuine RNA-binding events, aggregating the cross-linking data from proteins that share the same domains may provide valuable insights into preferred RNA-binding interfaces.

To test this hypothesis, we further analysed the cross-linking data for RRM-containing proteins. The RRM domains in which cross-linking was detected were structurally aligned using MM-align (Mukherjee & Zhang, 2009) and superimposed. For those RRM domains for which crystal structures were not available, AlphaFold2 structure models were used. Subsequently, the cross-linked peptides and amino acids were highlighted within the superimposed structures (Fig 5A and B). Typical RRM domains consist of four antiparallel β-sheets stacked on the top of two α-helices (Fig 5A).

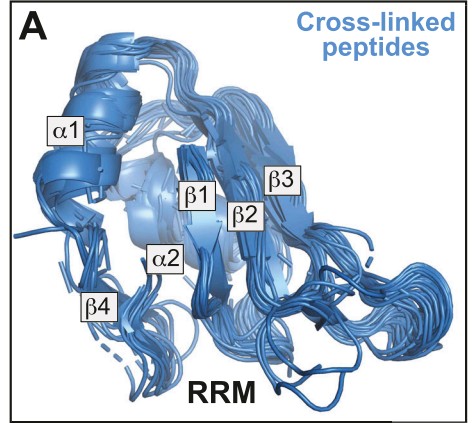

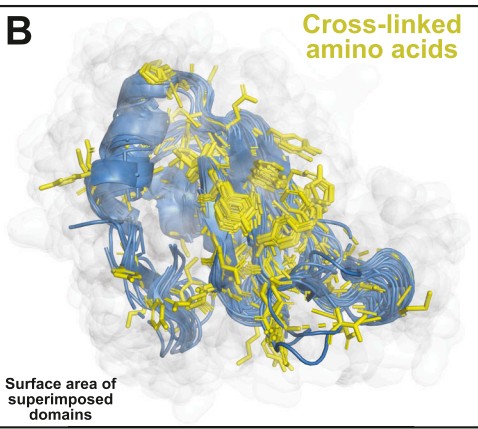

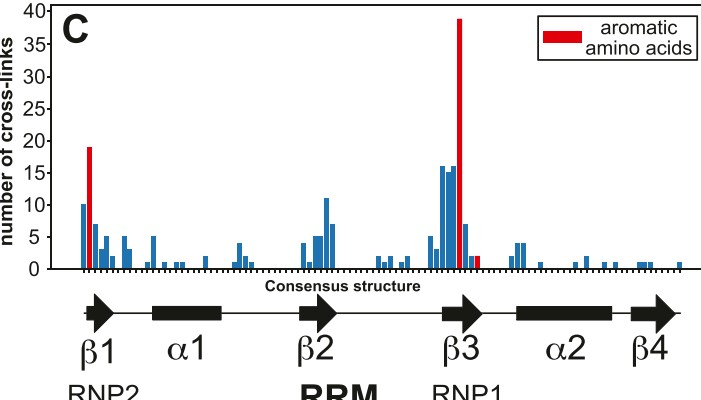

**Figure 5. Insights into RNA-binding interfaces in protein domains through aggregated amino acid UV cross-linking data.**
**(A)** Superimposed peptide sequences mapped to RNA recognition motif (RRM) domains in proteins identified in the RBS-ID dataset. These sequences were aligned on available structural models of RRM domain–containing proteins. The various α- and β-secondary structural elements within the RRM domains are also indicated. **(B)** As in (A), but with the side chains of UV cross-linking sites within the domains highlighted as yellow sticks. The white cloud represents the surface area of the RRM domains. **(C)** Number of UV cross-links detected in all superimposed RRM domains (y-axis), correlating to their specific positions within the domain (x-axis). Below the x-axis, the consensus secondary structure for RRM domains is depicted for reference.

Our analyses revealed that many cross-linked amino acids clustered in the same regions of the RRMs and concentrated in the β-sheets (Fig 5B). This finding is consistent with the essential role of the RRM β-sheets in RNA-binding (Maris et al, 2005). Moreover, aromatic amino acids from the first and third β-sheet that are important for RNA-binding (Maris et al, 2005) frequently cross-linked to RNA (Fig 5C, red bars). However, to obtain meaningful results, many cross-linking events within a specific domain are required. To illustrate this point, the same analyses on type 1 KH domain proteins (36 cross-links), which were also enriched in the RBS-ID data, did not reveal a convincing cross-linking pattern (Fig S5A and B). Nevertheless, our work demonstrates the potential of using high-throughput UV cross-linking studies for studying protein–RNA interfaces.

### Aggregating data from multiple predictors increases confidence in RBS identification

The pyRBDome data analysis pipeline was founded on the principle that integrating outcomes from various distinct predictors not only enhances the quality of RBDome data but also enables more reliable identification of RBSs in proteins for which cross-linking data are absent. These assumptions were tested using ML. Using the ground truth datasets outlined above, we developed eXtreme Gradient Boosting (XGBoost) ensemble classification models (Chen & Guestrin, 2016) that use the prediction results from the diverse tools used by pyRBDome as features to predict how likely an amino acid is to bind RNA (detailed in Figs 1 and S6). The XGBoost probability scores for spCas9, derived from all the prediction results generated for this protein, are shown in Fig 3G and displayed as the score bar in Fig S3.

Developing a robust ML model for predicting RBSs is challenging, requiring extensive benchmarking against existing tools and deeply curated ground truth datasets, which is beyond the scope of this study. However, precision–recall analyses (Fig 6A and B) indicated that the XGBoost classifiers trained on the combined prediction results of the human GT-Distance (Fig 6A) and GT-PLIP (Fig 6B) ground truth datasets exhibited much lower false-positive and false-negative rates compared with classifiers trained solely on data from individual tools, as shown by the substantially higher average precision (AP) of the XGBoost ensemble model. Furthermore, XGBoost models trained with more RBS prediction data also displayed much improved area under the curve (AUC) values (Fig 6C and D), implying they better distinguish between amino acids that bind RNA and those that do not.

Training XGBoost models using results from various combinations of RBS prediction tools revealed that models trained with a more extensive collection of prediction data showed increased precision (Fig 6E; AP). Notably, the AUC scores displayed less reliance on the number and type of RBS prediction datasets used (Fig 6F).

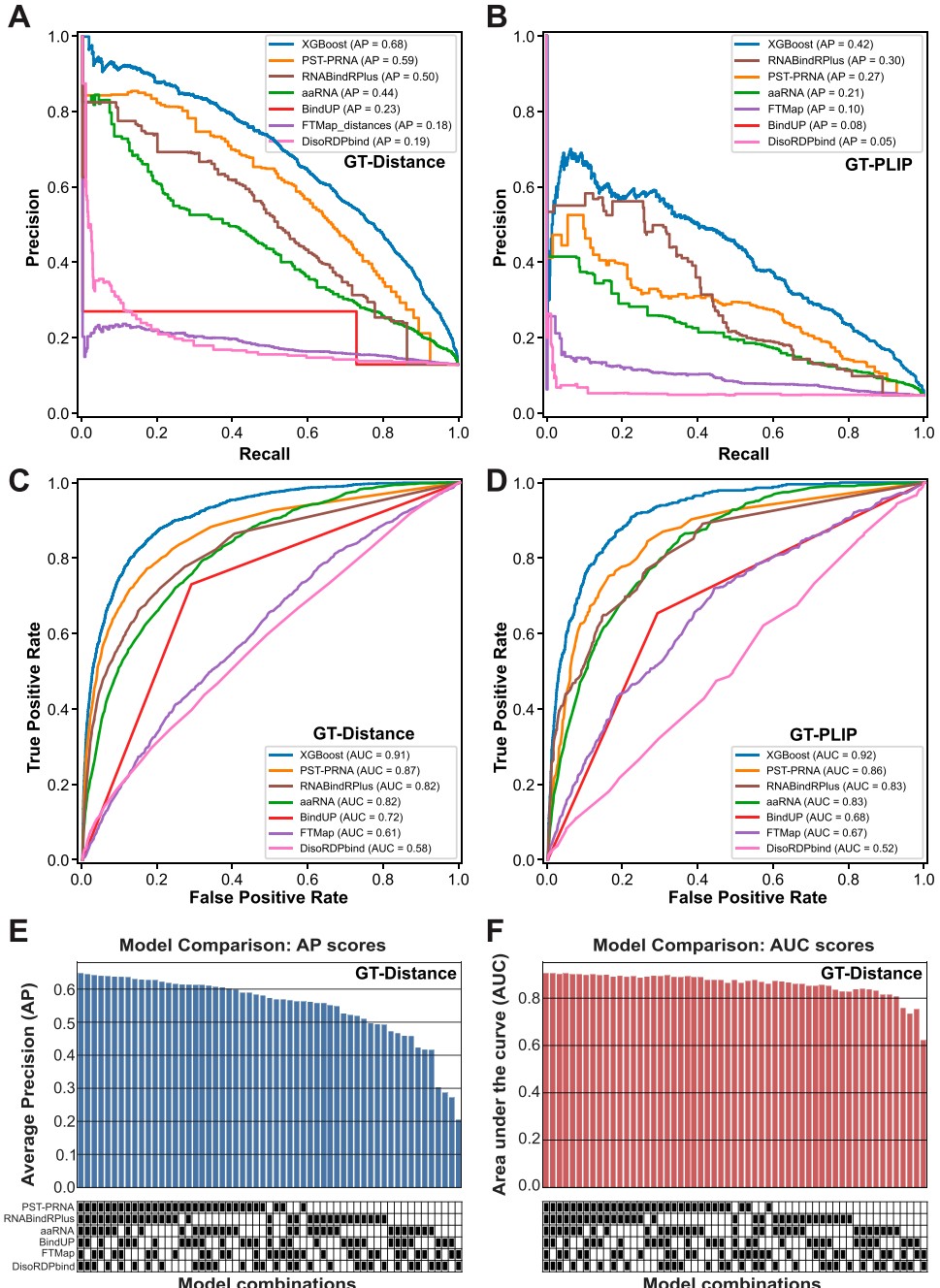

**Figure 6. Assessment of XGBoost models trained on prediction models.**
**(A, B)** Precision–recall curves for the various XGBoost prediction models trained on the GT-Distance (A) and GT-PLIP (B) ground truth datasets using the predictions from either the individual tools or all predictions combined. The AP score for each model is indicated in the legend (e.g., aaRNA AP = 0.44).
**(C, D)** Receiver operating characteristic curves for the same prediction models using the GT-Distance (C) and GT-PLIP ground truth datasets (D), with AUC scores provided in the legend. **(E, F)** Bar graph comparing the AP (E) and AUC (F) scores across different XGBoost models for the GT-Distance training dataset. The XGBoost models were trained on results from different combinations of prediction algorithms. The heatmap below the bar plot indicates what model combinations were used for training and testing the model.

We note that models trained on GT-PLIP generally performed poorer than might be expected. This is likely because not all available structures could be analysed by PLIP because of limited resolution, reducing the size of the training dataset. In addition, the unbalanced nature of the GT-PLIP dataset, with only ~5% of all amino acids interacting with RNA, likely also significantly contributed to the lower precision of the XGBoost models trained on the PLIP data, despite artificially balancing the datasets (see the Materials and Methods section).

Importantly, the individual RBS prediction tools (i.e., the model features) do not contribute equally to the predictions made by the XGBoost models, but the significance of each model is evaluated during the training. Analysis of the feature reliance in the performance of the XGBoost model (Fig S7A and B) revealed that PST-PRNA, RNABindRPlus, and BindUP exhibited the highest importance among the RBS prediction tools, enabling the model to approximate the ground truth more accurately.

Collectively, these results validate our premise that combining outputs from multiple tools can enhance the prediction of RBSs in proteins and establish a strong foundation for the development of more sophisticated ML models (see the Discussion section). In addition, these results highlight the flexibility of our XGBoost model: even if the user is unable to provide

results from some of the tools, the model can still generate predictions without substantially impacting its performance (Fig 6E). We subsequently used the XGBoost model trained on the GT-Distance data to predict RBSs in proteins from the RBS-ID data. All the results from these analyses are provided with the cross-linking information for each protein in Table S4. On our GitLab repository, we also provide PDB and PDF files summarising our XGBoost prediction results for all the proteins analysed during the project.

## pyRBDome correctly identifies RBSs in an *S. aureus* 3′–5′ exonuclease

We next applied the pipeline on RBPome data from a less well-characterised organism. We used our published RBPome data (Chu et al, 2022) generated on a clinically relevant *S. aureus* strain (USA300). This dataset was chosen as a representative example because it also contains many proteins with domains previously not associated with RNA-binding. Given that our current XGBoost model thus far had only been trained on human ground truth data, these analyses also tested the adaptability of the model to data from a genetically distant organism. Structures (from rcsb.org or AlphaFold2) for the top 200 enriched proteins were analysed by the pipeline, and the results are available on our GitLab repository (https://git.ecdf.ed.ac.uk/sgrannem/pyRBDome_Notebooks_Staphylococcus_aureus_analyses). To verify our findings, we focussed our analysis on the *S. aureus* polynucleotide phosphorylase (PNPase) 3′–5′ exonuclease, for which crystal structure data were available for both *S. aureus* (active site only) and *Caulobacter crescentus* (Hardwick et al, 2012; Wang et al, 2017). The latter structure also contained a short piece of RNA, enabling us to determine the agreement between the predictions and the structural data.

To obtain a structure with the complete *S. aureus* PNPase sequence, we downloaded the AlphaFold2 model as it was in good agreement with the published structures (root-mean-square deviation values between 0.6 and 1; Fig S8A). Some of the outputs of prediction results from individual predictors employed by pyRB-Dome are shown in Fig 7A.

PNPase consists of three subunits that form a ring-like central channel where the RNA threads through the enzyme (Fig S8B). The S1 and KH domains, located at the C-terminus of each subunit, form the entrance of the channel, and direct the single-stranded RNA towards the catalytic residues of the RNase PH-like domain, which is located at the N-terminal side of the channel (Hardwick et al, 2012). In the *C. crescentus* PNPase-RNA crystal structure, a 12-nucleotide RNA fragment interacts with the KH domain, through the conserved RNA-binding GSGG loop (Figs S8A and B and 7A–C). These amino acids were predicted to bind RNA with high probabilities by RNABindRPlus and our XGBoost model (Fig 7A–C). The predictions of our pipeline largely accumulated on the internal surface of the ring-like structure that interacts with RNA. This can easily be observed when overlaying the RNA from the *C. crescentus* structure on the pyRBDome PNPase structure with the model predictions highlighted in Fig 7B and C. Interestingly, although FTMap highlighted the PNPase active site for its high potential to bind small molecules (Fig 7A; red-coloured amino acids), this region showed

relatively low RNA-binding probabilities, reflecting the nuanced contribution of FTMap results to our XGBoost model's predictions (Figs 6 and S7). The aaRNA analysis on the PNPase model structure did not yield any results, and therefore, these data were missing when using our XGBoost model, which was trained with aaRNA data, for predicting RBSs in this structure. Despite this, the XGBoost model yielded correct predictions for PNPase-RNA-binding regions, again highlighting the degree of flexibility and robustness in the predictive capabilities of XGBoost models.

## UV-induced protein–RNA cross-links frequently occur in proximity to structurally determined protein–RNA contacts

We next asked to what extent the RBS-ID data agreed with our human ground truth datasets. For this purpose, we only considered UniProt IDs from the RBS-ID data for which protein–RNA structures were available. We then compared this selection of RBS-ID data with our PLIP-analysed structures (GT-PLIP dataset). For each cross-linked amino acid reported in the RBS-ID data, we measured the distance (in Å) to the nearest RBS detected by PLIP. The results were then aggregated into the cumulative plot shown in Fig 8A. Much to our surprise, these data showed that only 21.1% (43 of 204 amino acids) of the reported cross-linking sites interact with RNA in high-resolution structures (as reported by PLIP; Fig 8A). The previous work (Knörlein et al, 2022) demonstrated that UV does not necessarily always cross-link the amino acids that in available structures bind RNA, but neighbouring amino acids can also be indirectly covalently attached to RNA. Consistent with this idea, more than half (56.4%) of the cross-linked amino acids were located within hydrogen-bonding distance (4.2 Å) of PLIP sites and 42% within 4.2 Å distance of RNA in these structures (Fig 8B). Statistical analyses (Kolmogorov–Smirnov [KS] tests) revealed that RBS-ID data are indeed highly enriched for amino acid positions that are close to PLIP sites or RNA molecules in 3D structures (relative to shuffled cross-linked amino acids or all amino acids; Fig 8A and B). These data therefore reinforce the idea that UV cross-linking does not always capture amino acids *that in static protein–RNA structures* directly bind RNA, but that they are generally closer to these RNA molecules in these structures.

We next focussed specifically on the cross-linked amino acids that overlapped with RBSs in our GT-PLIP dataset and asked what type of interactions they are involved in. Consistent with the previous work (Knörlein et al, 2022), we find that phenylalanine π-stacking interactions with RNA are most abundantly detected (Fig 8C and D). However, our results also suggest important contributions to hydrophobic and π-cation interactions (Fig 8C).

## Cross-linked peptides as reliable proxies for RNA-binding regions?

As outlined above, a main reason why we established the pyRBDome pipeline was because for our model organism (methicillin-resistant *S. aureus*), there were an insufficient number of high-resolution structures of protein–RNA complexes available to generate a robust ground truth dataset. When analysing data from less well-

**A**

**B**

*C. crescentus* PNPase
PDB ID 4AM3

**C**

pyRBDome model
predictions

**Figure 7. pyRBDome detects known RNA-binding regions in *S. aureus* polynucleotide phosphorylase (PNPase).**
**(A)** Results from prediction algorithms on the surface representation of a PNPase monomer. The colours for BindUP, DisoRDPbind, and RNABindRPlus results indicate RNA-binding probabilities, with cooler shades (blue) suggesting lower and warmer shades (red) indicating a higher RNA-binding likelihood. For the FTMap results, warmer red shades signify shorter distances to docked molecules. The active site of the nuclease is marked with a square box. The GSGG loop is marked with a red square box. Blue colours represent amino acids with low RNA-binding prediction scores (BindUP, DisoRDPbind, or RNABindRPlus), whereas red colours indicate amino acids with high RNA-binding prediction scores. For the FTMap data, the blue-to-red colour gradient denotes decreasing distance to docked small molecules, with red indicating distances of ≤2 Å and blue indicating distances of >4.2 Å. Accompanying colour bars represent the RNA-binding propensities, correlating specific colours with their respective values. **(B)** Crystal structure of PNPase from *C. crescentus*, in complex with RNA, PDB ID 4AM3 (Hardwick et al, 2012). The RNase PH-like domains, coloured in dark and light pink, are linked by a helical domain, coloured in yellow. The KH domain (green) interacts with the RNA of the structure through the GSGG loop (red). The S1 domain is absent from this crystal structure. **(C)** Structural alignment of the RNA from structure 4AM3 on the PNPase AlphaFold2 model with results from XGBoost model predictions trained on the prediction results from all algorithms. Catalytic residues are displayed as spheres and are highlighted in an enlarged view of the active site region.

characterised organisms, the user can instruct the pipeline to determine whether cross-linked peptides and/or amino acids are highly enriched for RBSs predicted by the various tools employed by pyRBDome. In addition, the user can test whether the cross-linking data are enriched for amino acids that, according to our XGBoost model, have high RNA-binding probabilities. Examples of such analyses on the human RBS-ID data are shown in Fig 9A and B. These data indicate that the reported cross-linked amino acids have a significantly higher likelihood to bind RNA compared with randomly selected amino acids from the same proteins or the general population of all amino acids from the analysed proteins. However, the variability in the distribution of the RNA-binding probabilities for cross-linked RNAs, as shown by the lower tail of the distribution, indicates that although cross-linked amino acids are indeed more likely to be predicted as RNA-binding, they are not a definitive indicator by itself.

Therefore, we next asked whether cross-linked *peptides* might be a better proxy for RBS detection. The pyRBDome pipeline allows the user to test this in two ways: Firstly, the user can compare the data with results obtained from individual predictors, aaRNA or FTMap, for example, as illustrated in Fig S9. These data show that the

generated RBS-ID peptides were both enriched for predicted RBSs and more likely to be in closer proximity to these sites (aaRNA and RNABindRPlus; Fig S9A and B). Interestingly, the same was true for putative small molecule binding sites predicted by FTMap (Fig S9C). The second approach determines whether the cross-linked peptides are enriched for amino acids with higher RNA-binding probabilities as determined by our XGBoost model. We addressed this by (I) tracking the highest RNA-binding probability found in a peptide sequence (Fig 9B) and (II) calculating the mean RNA-binding probabilities for each peptide (Fig 9C). Our analyses strongly indicate that cross-linked peptides typically include at least one amino acid with a significantly higher RNA-binding propensity compared with control samples (Fig 9B). Notably, the RNA-binding probability distribution shown in Fig 9B for cross-linked peptides is distinctly skewed towards higher values, suggesting that these peptides have a greater tendency for containing RBS relative to the randomly selected control group peptides. However, the randomly generated peptides were not products of trypsin and/or Lys-C digestion. To address this, we also compared the cross-linking data with peptides from parent proteins digested in silico by trypsin/Lys-C. This comparison showed an even higher

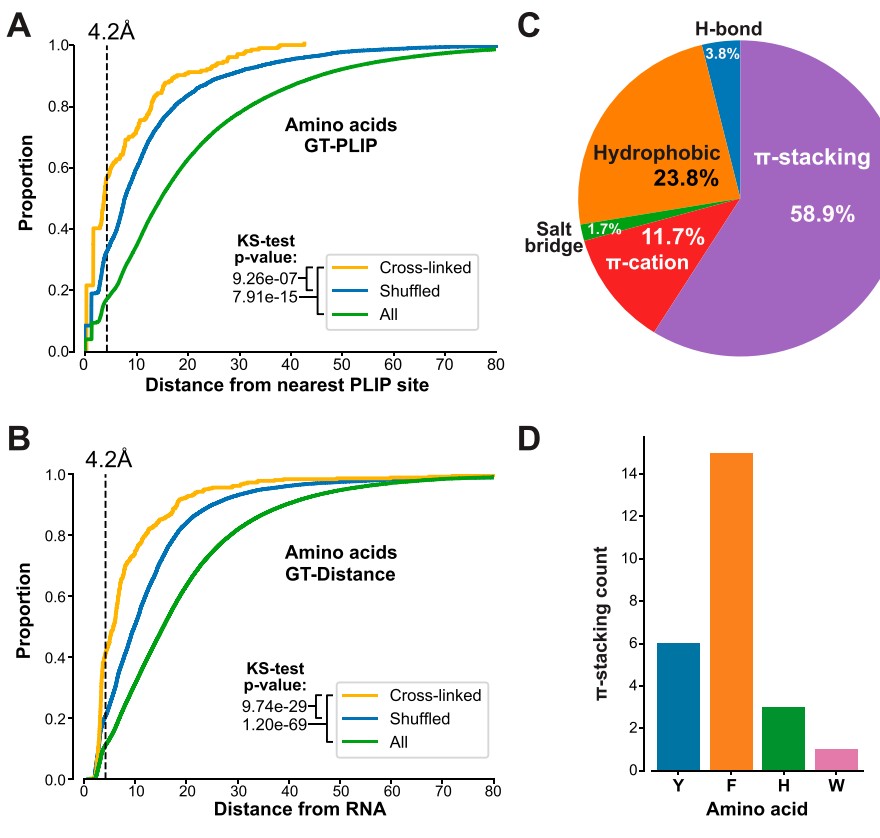

**Figure 8. Limited concordance between UV cross-linking data and protein–RNA structures.**
**(A)** Cumulative distribution of distances for cross-linked amino acids (yellow), randomly shuffled amino acids (blue), and the total pool of amino acids (green), in comparison with established RNA-binding amino acids determined by protein–ligand interaction profiler. *P*-values, calculated using the KS test, indicate significant differences between groups. The 4.2 Å threshold, indicated by the dashed vertical line, is used to determine the proximity required for hydrogen bonding. **(B)** Similar to (A), this analysis plots the cumulative distances of cross-linked, randomly selected, and all amino acids within the studied RNA-binding proteins, relative to their proximity to RNA. The KS test was also employed here to calculate *P*-values. **(C)** Amino acids that form π-stacking interactions are often cross-linked to RNA. The pie chart displays the percentages of each cross-linked amino acid involved in different types of interactions: hydrogen-bonding (H-bond), π-stacking, π-cation, salt bridge, and hydrophobic interactions, as identified by protein–ligand interaction profiler. These percentages were calculated by dividing the number of a specific type of interaction by the total number of such interactions detected in the analysed structures. **(D)** Counts of cross-linked amino acids involved in p-stacking interactions. Y = tyrosine; H = histidine; F = phenylalanine; and W = tryptophan.

presence of predicted RBSs in cross-linked peptides, affirming the predictive strength of our XGBoost model and the significant value of cross-linked peptide data for detecting RBSs.

Therefore, when analysing cross-linked peptides from RBDome data using the pyRBDome pipeline, we advise users to prioritise further characterisation of cross-linked peptides that contain positively charged and/or aromatic residues with high RNA-binding propensities, as predicted by XGBoost.

## pyRBDome: a viable computational alternative for experimental RBDome methods?

Given the technical challenges associated with performing RBDome experiments and analysing the resulting data, we asked whether pyRBDome could be a viable alternative approach for detecting the RBS in RBPs. To assess this, we calculated performance metrics (accuracy, precision, recall, F1 score, and Matthews correlation coefficient) for both the XGBoost predictions and the cross-linking data on the spCas9 and the proteome datasets. Given that the spCas9 structure was not used for training, we could employ our existing XGBoost model to predict putative RBS in this protein. To evaluate the performance metrics for the proteome RBS-ID data, we trained a new XGBoost model so that we could make predictions for a larger number of proteins in the RBS-ID data (Fig 10A and B; proteome data). We then examined how the predictions and the cross-linking data agreed with the structural data, considering amino acids that bind RNA according to the

structural data (GT-PLIP; Fig 10A) or those within hydrogen-bonding distance of RNA (GT-Distance; Fig 10B) as RBS. The results demonstrate that the XGBoost model substantially outperforms the experimental cross-linking approach across nearly all metrics, suggesting more reliable and accurate predictions of RBS. The model's strong performance on the spCas9 dataset, evidenced by the moderate-to-high Matthews correlation coefficient values, is particularly noteworthy. Although the relatively low precision of the XGBoost predictions indicates a higher rate of false positives, this is not uncommon given the high recall rate. It is, however, important to note that cross-linking is stochastic and inefficient and does not necessarily happen uniformly or predictably on all RBSs. These two factors contribute significantly to the low recall observed in the cross-linking data.

Given the limitations of cross-linking and RBDome data, these results strongly support a role of pyRBDome in enhancing the experimental data to generate more reliable overviews of protein–RNA interaction sites. Although the results from the XGBoost model are encouraging, it is important to note that the model has been trained predominantly on known folds and canonical RNA-binding domains. Consequently, it remains unclear how the model performs on atypical RNA-binding domains. However, statistically evaluating the model's performance in these unusual domains is currently very challenging given the scarcity of mechanistic insights into how these domains interact with RNA. This represents a limitation in the current study that will be addressed once more data from diverse RBPs and RNA-binding domains become available.

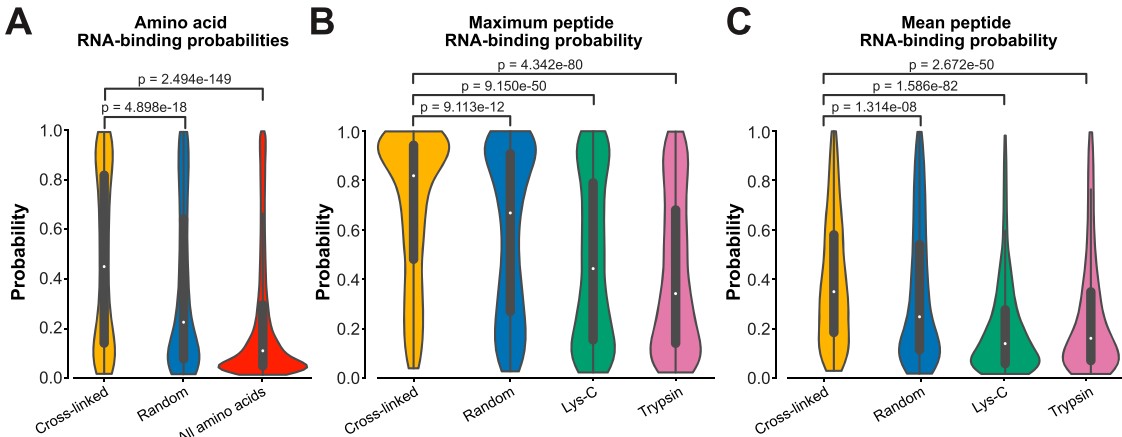

**Figure 9. Cross-linked peptides as reliable proxies for RNA-binding sites.**
**(A)** Violin plots showing the distribution of RNA-binding probabilities as determined by our XGBoost model for cross-linked, randomly shuffled amino acids, and all available amino acids within the analysed RNA-binding proteins. **(B)** Distribution of the highest RNA-binding probability score (determined by our XGBoost models) detected in cross-linked peptide sequences. Control datasets included randomly generated peptides with the same length distribution, and peptide libraries generated in silico by Lys-C or trypsin digestion of the RNA-binding proteins analysed here. **(C)** As in (B), but now for the average RNA-binding probabilities calculated for each cross-linked peptide. *P*-values, calculated using a two-sided Mann–Whitney–Wilcoxon test with the Bonferroni correction, indicate significant differences between groups, as shown above each comparison. The violins represent density estimations of the distances, with wider sections indicating a higher frequency of distances. The white dot in the centre of each violin plot denotes the median distance, and the thick lines within the violins represent the interquartile ranges.

## Discussion

Here, we present the pyRBDome pipeline for in silico enhancement of RBPome and RBDome proteomics data. This pipeline, which leverages both protein sequences and structural information, employs a variety of distinct prediction tools for identifying putative RBSs within target proteins (Figs 1, S1, and S6). It subsequently highlights the results from each prediction algorithm within either provided peptide/amino acid sequences or entire protein sequences. The pipeline is capable of processing hundreds of proteins from large proteomics datasets or individual proteins. Significantly, the pipeline simplifies the complex data from these predictions, providing easily interpretable results that facilitate identification of residues involved in RNA-binding. The inclusion of PyMOL sessions allows users to visualise all the experimental and prediction results in 3D model structures simultaneously. Furthermore, pyRBDome includes statistical analyses to assess whether sequences obtained from RBDome studies show significant enrichment of predicted RBSs, thus offering a quantitative measure that can improve the quality of the experimental data. Collectively, these findings underscore pyRBDome's utility in streamlining the detection of RBSs in proteins and in effectively enhancing RBDome data.

### Agreement between RBDome UV cross-linking and structural data

To demonstrate the utility of pyRBDome, we analysed a data-rich human RBDome dataset (RBS-ID; Bae et al, 2020), which provided, besides a list of (putative) RBPs, also an extensive list of RNA–cross-linked amino acids. However, it did not contain the peptide sequences to which these cross-linked amino acids belonged. To address this, we artificially extended these amino acid sequences on

both ends with varying lengths to create a peptide dataset suitable for analysis with our pipeline. We found that both cross-linked peptides and amino acid sequences are significantly enriched in RBSs, as predicted by individual tools or our combined XGBoost ensemble model. Surprisingly, when we compared the cross-linked amino acid data to our GT-PLIP dataset, which includes amino acids known to interact with RNA based on structural data, we observed a limited overlap. Although cross-linked amino acids were statistically more likely to be near RNA compared with randomly selected amino acids, only about 21% of them were found to bind RNA according to the available structural data. The limited overlap observed might suggest that UV cross-linking data contain a considerable amount of noise. However, it is important to note that our ground truth datasets, which were constructed solely from high-resolution structures, are also unlikely to include all possible protein–RNA contacts. Many structures contain proteins in complex with short pieces of RNA and therefore provide limited insights into the full RNA-binding capacity of the protein. Not every RNA substrate will also interact identically with an RBP, and protein–RNA interactions can be highly dynamic and condition-dependent. Though UV cross-linking can often capture such interactions in vivo and in cellulo, many of these might not be represented in static structures (also see Bae et al, 2021).

Our comparison of the RBS-ID data with our XGBoost model predictions suggests that sequences of cross-linked peptides are more reliable indicators of RBSs than individual amino acids. This is because they tend to include amino acids with higher RNA-binding probabilities. Thus, comparing the cross-linking data with results from predictive models may offer a more effective solution for corroborating or supporting RBDome data. This is particularly true for models that are not solely reliant on existing protein–RNA structures for training. Such models are presumably better equipped to identify amino acids interacting with RNA, including those interactions not represented in structural data.

## A  GT-PLIP

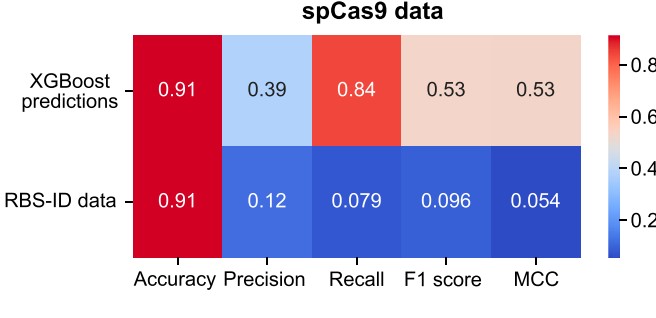

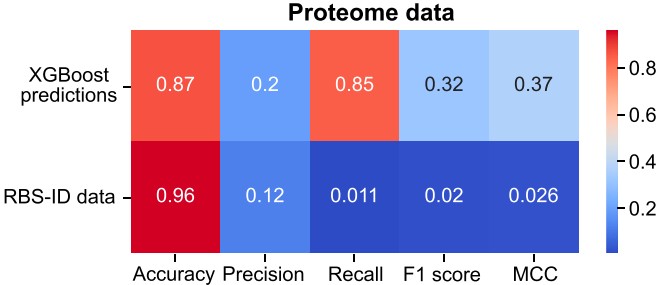

## B  GT-Distance

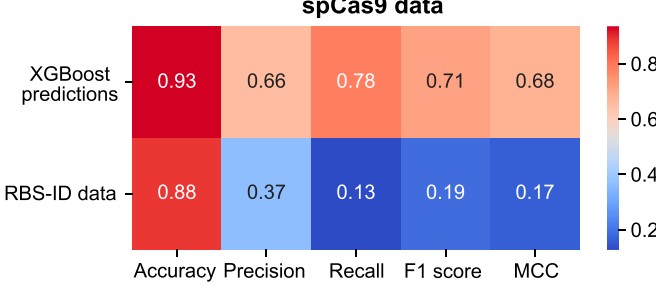

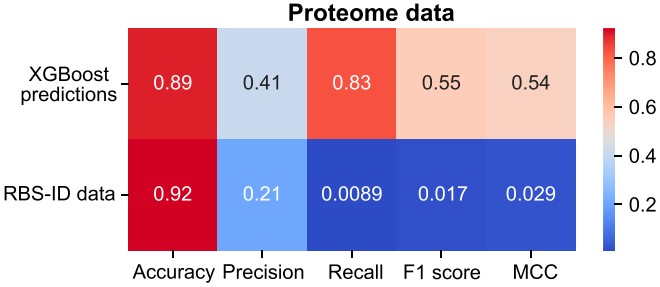

**Figure 10.  Performance metrics used when comparing the RBS-ID cross-linked amino acids to the XGBoost predictions and ground truth datasets.**
Colours closer to red indicate high performance, whereas colours closer to blue indicate poor performance. **(A)** Comparing the XGBoost predictions with the spCas9 and proteome RBS-ID cross-linking data using GT–protein–ligand interaction profiler data generated from published protein–RNA structures. **(B)** Same as in (A) but now considering amino acids that are within 4.2 Å from RNA as RNA-binding residues. Explanation of the performance metrics: **accuracy**: the frequency of model predictions that are correct. Note that mostly the non–RNA-binding amino acid residues are correctly predicted. **Precision**: measures what fraction of the items predicted as positive by RBS-ID and XGBoost are also positive in the ground truth datasets. **Recall**: measures what fraction of the positive items in the ground truth datasets were identified correctly by RBS-ID and our XGBoost model. **F1 score**: the harmonic mean of precision and recall. The F1 score reaches its best value at 1 (perfect precision and recall)

Another potential source of noise could stem from the analysis of MS data. The software tools employed for analysing such datasets typically offer localisation scores, which indicate the probability of an amino acid being cross-linked to RNA. If the quality of a dataset is subpar, accurately pinpointing the precise cross-linking site becomes more challenging, leading to lower localisation scores and, consequently, increased noise in the data. However, in the RBS-ID dataset that we analysed (Bae et al, 2020), 80% of the reported cross-linking sites (detected using MS-GF+ with a closed search [Kim & Pevzner, 2014; Bae et al, 2020]) had very high localisation scores (between 0.8 and 1). Although there is undoubtedly noise in the data, we would argue that the quality of this RBS-ID dataset is not a major contributor.

A recent study has also revealed that UV cross-linking does not exclusively target amino acids in direct contact with RNA; it can also affect those in indirect proximity (Knörlein et al, 2022). Furthermore, it was proposed that $\pi$-stacking interactions are key to directing the cross-linking reactions (Knörlein et al, 2022). This may also explain our observation that few cross-linked amino acids were found to bind RNA in our GT-PLIP ground truth dataset, and if they did, they were mostly involved in $\pi$-stacking. However, a significant proportion of the cross-linked amino acids were observed to be near RNA within protein–RNA structures. Drawing on these findings and the bioinformatics analyses conducted in this study, when using pyRBDome data to design follow-up mutational analyses, we recommend prioritising aromatic and positively charged amino acids that have high RNA-binding prediction scores, that have undergone cross-linking or are in cross-linked peptides, and that are proximal to cross-linking sites, either sequentially or in the three-dimensional (model) structures.

### Developing an ensemble model for enhanced prediction of RNA-binding amino acids

The foundational concept behind the creation of the pyRBDome pipeline stemmed from our belief that combining results from multiple predictors would improve the identification of RBSs in targeted proteins. Although this was not the main focus of our project, the comprehensive datasets generated by pyRBDome presented a prime opportunity to validate this hypothesis through ML. By leveraging the predictive data from various tools, we developed XGBoost ensemble models. These models discern patterns within the aggregated predictive results and align them with known RBSs in the existing structural data. The main reasons for relying on XGBoost to build these preliminary models include its frequent outperformance of neural networks when presented with tabular data (such as the data used here; Fig S6), its ability to handle missing data points effectively (useful in cases where a protein could not be analysed by one of the prediction tools), its competence in dealing with unbalanced datasets (our ground truth datasets are unbalanced), and its tolerance to uninformative

---

and worst at 0. **MCC**: the Matthews correlation coefficient considers true and false positives and negatives and returns a value between –1 and +1, where +1 indicates perfect prediction, 0 indicates random prediction, and –1 indicates total disagreement between prediction and observation.

features (Chen & Guestrin, 2016; Grinsztajn et al, 2022 *Preprint*). XGBoost therefore provided an excellent starting point for developing improved models for RBS prediction.

The preliminary models we constructed substantially outperformed the individual tools, demonstrating greater accuracy and precision in predictions (Fig 6). Although these results are promising, there are areas where the XGBoost models could be further improved. For instance, our current models have exclusively been trained on data from human protein–RNA complexes. Therefore, their robustness could be enhanced by training the models on structurally characterised protein–RNA complexes (RNPs) from diverse organisms. It should also be noted that our training sets, in addition to AlphaFold2 models, mainly consist of structurally characterised proteins/domains. As a result, RBPs with disordered RBSs are underrepresented or their disordered regions were excluded from the analyses. This underrepresentation likely contributed to the less optimal performance of DisoRDPbind on our test data. However, this can be improved by reanalysing the data using *only* AlphaFold2 models, where these sequences will be represented (albeit not accurately folded). Alternatively, including a wider array of RNA-binding domains from disordered regions (Zhang et al, 2022, 2023) will undoubtedly enhance DisoRDPbind's predictions and subsequently further improve the accuracy and precision of our XGBoost models. Therefore, the analyses presented here, constrained by the current datasets, do not fully capture the true potential of DisoRDPbind.

## Pipeline performance

The pyRBDome pipeline was designed to process many proteins simultaneously, naturally leading to questions about the typical duration of an RBPome or RBDome dataset analysis. Although there is no definitive answer, as it varies, performing the pyRBDome analysis on the RBS-ID dataset (consisting of 584 proteins) took ~8 d. The most time-consuming step involved submitting jobs to various servers, with tools such as FTMap and aaRNA typically taking longer to yield results. The analysis duration primarily depends on factors such as the size of the proteins being analysed, the server's computational power, and the server queue lengths. Despite these variables, we consider an 8 d turnaround to be quite reasonable for such a large dataset. Future developments of pyRBDome, as discussed in the next section, will focus on incorporating tools with shorter execution times. However, it is important to note that faster processing does not always equate to more accurate results, presenting a constant trade-off.

## Future pyRBDome pipeline developments

To develop the pyRBDome pipeline, we evaluated a wide array of distinct tools designed to predict RBSs and that take into consideration various sequences and structural features of ligand-binding proteins. However, integrating these tools into pyRBDome presented several challenges. These included inactive web servers (such as the aaRNA web server shutting down) and compatibility issues such as dependency conflicts and lack of comprehensive

documentation, which hindered smooth integration with our Linux servers. Moreover, not all the web servers we tested were suitable for high-throughput analysis of protein sequences and structures, and some had run times that made the analysis of hundreds of proteins excessively time-consuming. This presented a notable challenge in integrating tools that could potentially outperform those currently described. However, the pipeline is continually evolving, and our existing Python code allows for relatively straightforward incorporation of new tools and the processing of their results.

Throughout this project, numerous advancements have been made in developing improved methods for predicting RBSs in proteins. A notable example is DeepDISOBind, an improved model for predicting RBSs in disordered regions (Zhang et al, 2022). We are in the process of incorporating the stand-alone version of this tool into pyRBDome-Core and pyRBDome-Notebooks. Other tools under evaluation are NCBRPred (Zhang et al, 2021), a sequence-based predictor likely to replace RNA-BindRPlus, and HybridRNAbind, a tool trained on both structural information and available RBSs in disordered domains (Zhang et al, 2023). We also tested HydRA (Jin et al, 2023), a deep learning method designed for detecting RBPs and RBSs. Like the XGBoost model described here, HydRA functions as an ensemble classifier, using information from diverse prediction tools. It not only predicts a protein's RNA-binding capacity but can also detect potential RBSs in RBPs. Using our human GT-PLIP and GT-Distance ground truth datasets, HydRA's performance in detecting RBSs was not as high compared with the individual tools employed by the pyRBDome pipeline or our XGBoost ensemble model (see 6.1.2_BinaryClassifierAnalysesRBDData.ipynb notebook in the pyRBDome-Notebooks Ground truth analyses repository). This is why we do not discuss the HydRA results here. This may be due to HydRA being optimised for predicting RNA-binding *regions*, whereas our ground truth datasets are more specific to individual RNA-binding *amino acids*. Despite this, we recognise HydRA's value in identifying RNA-binding capacities in proteins and have incorporated code in version 0.2.0 of pyRBDome-Core and version 1.1 of pyRBDome-Notebooks to process and display HydRA predictions in PDF and PDB files. All the raw HydRA analysis results are also available on our pyRBDome-Notebooks GitLab repositories.

Another enhancement that could be included in the pyRBDome pipeline is to incorporate tools that predict the RNA sequences that RNA-binding domains interact with (reviewed in Arora and Sanguinetti [2022a]). For example, the deep learning model NucleicNet (Lam et al, 2019) can predict RBP RNA-binding preferences, including RNA backbone and bases. Similarly, when providing protein and RNA sequences, GraphProt (Maticzka et al, 2014) and RBPsuite (Pan et al, 2020) make it possible to identify possible RNA sequences and structural preferences for RBPs. New tools implementing graph convolutional networks (Arora & Sanguinetti, 2022b) can learn RNA-binding preference for RBPs throughout entire transcriptomes, making it possible to build interaction networks for an RBP that are adaptable to different environmental conditions.

One might argue that constructing a pipeline dependent on multiple web servers, as in the case of pyRBDome, inherently invites

reliability issues, as demonstrated by our experiences with inconsistent server availability. Although our efforts are increasingly directed towards integrating stand-alone packages into the pyRBDome pipeline, it is important to acknowledge that running these prediction algorithms demands substantial computational resources. This includes the need for high-specification CPUs (central processing units) and, more critically, GPUs (graphics processing units). Not all research groups may have access to such computational facilities. Moreover, even for groups that do have such resources, the task of establishing and managing a pipeline comprising various stand-alone ML tools is very challenging as it involves dealing with numerous dependencies and configurations. Therefore, for future versions of the pyRBDome pipeline, we aim to strike a balance between using web servers and integrating stand-alone packages.

A longer term goal is to make the results from analyses available in public databases with the aim of making the data more easily accessible and for the wider public.

# Materials and Methods

## Repository content

A description of all the directories and type of files that the pyRBDome pipelines produce can be found in the README.md files in the individual repositories. The analyses described here used code from pyRBDome-Core versions 0.2.3 and 0.2.4, pyRBDome-Notebooks version 1.0, and pyRBDome-Notebooks Ground truth analyses version 1.1.2 and 1.1.5.

## Generating the human ground truth dataset

We used the UniProt IDs from the RBS-ID dataset (Bae et al, 2020) to search rcsb.org for available protein–RNA structures. To expedite this process, we developed the script FindUniProtRNPStructure.py, which is now part of the pyRBDome-Core package. The code used for downloading these PDB files is available in the 1.0_FindRNP-Structures_using_UniProt_IDs.ipynb notebook, located in the pyRBDome-Notebooks Ground truth analyses repository (https://git.ecdf.ed.ac.uk/sgrannem/pyRBDome_Notebooks_Ground_truth_analyses). For each UniProt ID, we retrieved protein–RNA structures that met specific criteria: a resolution of less or equal to 5 Å and the presence of at least one RNA molecule. Owing to compatibility issues with CIF files, we chose to download only PDB files from rcsb.org. Each PDB file corresponding to a UniProt ID was then analysed to determine the minimum distance (in Å) of each amino acid to the RNA. We also developed a Python package that uses the PLIP code (Adasme et al, 2021) to identify amino acids that interact directly with RNA in these structures. The code for conducting these analyses and a description of how to carry out such analyses are provided in the pyDRBPNA package on our repository (https://git.ecdf.ed.ac.uk/sgrannem/pyDRBPNA).

To further refine these ground truth datasets, we merged the distance calculations and PLIP results for all PDB files associated with a single UniProt ID into a composite PDB file. This file records only the shortest distances to RNA for each amino acid in the

b-factor column, as indicated in files ending with "distances_merged.pdb." We also collated the frequency of RNA contacts by amino acids across the structures (as detected by PLIP) and stored this information in the b-factor columns of files that end with "plip_merged_all.pdb."

## pyRBDome package and pipeline description

The pipeline introduced in this study consists of two parts: pyRBDome-Core (https://git.ecdf.ed.ac.uk/sgrannem/pyRBDome_Core) and pyRBDome-Notebooks (https://git.ecdf.ed.ac.uk/sgrannem/pyRBDome_Notebooks). The former contains all the scripts, functions, and classes that users need to execute the Jupyter notebooks. The code has been developed and tested extensively on Ubuntu Linux operating systems (OS) and can be adapted to work on Mac OS (12.7 and above). Details on how to install the packages and run the notebooks, and the required computational resources can be found in the README files on our repository. pyRBDome-Notebooks streamlines the process of the RBP and cross-linking data analysis by automatically running predictions either online or locally. It then downloads, renames, and organises the results into specific directories. The pipeline stores any progress it has made, as well as result from all the analyses in a SQLite database. This enables the user to keep track for which proteins (model) structures have been downloaded and whether these structures were analysed successfully by each prediction algorithm. Incorporating the SQLite database also enables the user to resume runs that may have failed or timed out and helps avoid repeated submission of PDB files that have already been analysed. The results tables can also be easily exported to CSV files. All the notebooks can also be run sequentially in the terminal using papermill (https://papermill.readthedocs.io). Papermill is automatically installed when installing the pyRBDome-Core package.

The pyRBDome-Notebooks Jupyter notebooks each have their unique number. A detailed description of what analyses each notebook does is outlined below.

### Finding all available (model) structures for each UniProt ID
pyRBDome-Notebooks notebook 1.0_FindingPDBs.ipynb was used to download all available PDB files (≤5 Å resolution) associated with the UniProt IDs listed in the RBS-ID data (Bae et al, 2020) from rcsb.org (Berman et al, 2000), model structures that were generated by AlphaFold2 (Jumper et al, 2021) or the SWISS-MODEL web server (Guex et al, 2009; Bienert et al, 2017; Waterhouse et al, 2018; Studer et al, 2020). For generating model structures, this notebook first queries the AlphaFold2 database (https://alphafold.ebi.ac.uk) and downloads the latest model associated with that UniProt ID (PDB files ending with "_AF.pdb"). If it is unable to find any models, it submits the protein sequence to SWISS-MODEL. Only models with a GMQE score higher than 0.7 were considered and their PDB files downloaded. Note that SWISS-MODELS were not used in this study and this functionality will be removed in future pyRBDome versions. When models are not available for proteins of interest, the user can provide their own models or structures in combination with notebook 1.1 to run them through the pipeline. This is how we analysed the spCas9 data using PDB ID 4un3 (Anders et al, 2014).

These models need to be added to a folder called "analysed_pdbs" in the same directory as where the pyRBDome notebooks are stored. The names of these PDB files also need to be provided in a CSV file called "protein_list.csv." Shown below is the content of the protein_list.csv files used for analysing the spCas9 data:

number, ID, name, pdb_id—1, Q99ZW2, spCas9, 4un3

The PDB IDs associated with each protein are then saved in the available_PDBs table in a SQLite database (pyrbdome_full.db). The tables in the database have information about whether the PDB file was successfully downloaded and what chain is included in the PDB file.

All the code used for analysing the RBD-ID spCas9 data and the analysis results are available from our repository (https://git.ecdf.ed.ac.uk/sgrannem/pyrbdome-notebooks-spcas9-analysis-results).

### Getting protein domains from Pfam
After all the PDB files have been downloaded, notebook 1.3 will use the InterProScan tool (Jones et al, 2014; Blum et al, 2021) to download all the domain information associated with these proteins. Only Pfam domains are considered. A Linux version of InterProScan is provided in pyRBDome-Notebooks programs folder. The user will need to install a different version if Mac OS are used for the analyses.

### Creating peptide control datasets
Notebook 1.3 takes the protein sequence from each PDB file and digests the sequences in silico with trypsin and Lys-C to generate a library of all possible peptides that could theoretically be detected by the mass spectrometer for the protein of interest. If cross-linked peptide sequences were provided, notebook 1.4 will generate a library of random peptide sequences that are peptides of the exact same length distribution as the cross-linked peptides, but that were randomly extracted from the protein sequence.

### Performing RNA/ligand-binding site predictions
To predict RNA/ligand-binding sites on the proteins of study, we chose six different prediction algorithms: aaRNA, PST-PRNA, BindUP, FTMap, RNABindRPlus, and DisoRDPbind (Mehio et al, 2010; Walia et al, 2014; Peng & Kurgan, 2015; Paz et al, 2016; Li & Liu, 2022) (see Table 1 for an overview of the tools). The series 2 notebooks will automatically submit all the PDB files to the respective web servers, download the results, and store the progress they have made with the analyses in the SQLite database. Note that the aaRNA server is no longer online, and therefore, this tool will be removed from future versions of pyRBDome.

### Mapping the cross-linked amino acid and peptide sequences to the PDB files
Notebook 3.0 takes the cross-linked, in silico–digested, and random peptide sequences and maps them to the PDB files. Once the peptides have been mapped, it will determine the location of cross-linked amino acids, if this information was provided. For example, if the peptide sequence "PSRKDPKYREWHHFL" is analysed by this notebook and it could be mapped to a PDB file sequence, it will record the start and end residue numbers for the peptide and what chain it was mapped to in the PDB file. For this example, the code returned the following result: 74A_psrkdpkyrewhhfl_88A. This shows that the peptide was mapped between residues 74 and 88 of chain A in the PDB file. Note that not all peptides will be mapped as many structures do not contain the complete protein sequence.

### Processing the results and storing them in PDB files
Notebook 4.0 collects all prediction results and any domain and mapped peptide/amino acid information and stores the results in the b-factor columns of the PDB file. This makes it possible to visualise the results in PyMOL or other viewers.

### Distance analyses
The series 5 notebooks take all the prediction results, map these to the peptide sequences, and calculate the closest distance of the cross-linked peptides or control peptide sequences to amino acids predicted to be involved in RNA-binding. The results are stored in tables in the SQLite database. These tables enable the user to easily extract peptide sequences that contain predicted RBSs. For example, if it found a predicted RBS in a mapped peptide (e.g., 74A_psrkdpkyrewhhfl_88A), it will indicate the location of this amino acid in upper case (e.g.,74A_psrkdpky**R**ewhhfl_88A).

### Sanity check
Notebook 6.0 then looks at all the distance analyses and double-checks that no errors were made in the calculations. This notebook is tremendously useful for troubleshooting any issues that might appear during the analysis.

### Making the final output files
Notebooks 6.1–6.4 gather all the prediction and cross-linking information from the PDB files that were produced by notebook 4 and place the information in a large table where RNA-binding probabilities provided by each algorithm are stored, as well as the location of cross-linked peptides and amino acid residues. The notebooks in the pyRBDome-Notebooks analyses of the ground truth dataset also contain extra code that adds the distances to RNA molecules for each amino acid for all protein–RNA structures that were analysed. Notebooks 6.1 and 6.2 take all the prediction results available in the large table, feed that to our XGBoost models, and calculate for each amino acid in each protein a probability for RNA-binding. Notebook 6.3 generates PyMOL session files that enable the user to conveniently load all PDB files into a single PyMOL session. Notebook 6.4 summarises all the results in the protein sequence (see Fig S3 as an example). The score bars in the PDF files indicate the XGBoost RNA-binding probabilities for each amino acid.

### Analysis of cross-linked peptide and amino acid sequences
The series 7 notebooks search for enriched tripeptide motifs enriched in the cross-linked peptides and enriched amino acids in the cross-linked amino acid data, if available. It returns a table containing the sequences of the enriched amino acid motifs or chemical properties and associated P-values. The 8.2 statistical analysis notebook determines whether cross-linked peptides and amino acids (where available) are significantly enriched for predicted RBSs compared with the random peptide datasets and the peptides generated by trypsin/Lys-C digestion of the protein sequences.

### Binary classification analyses: training of XGBoost models

The pyRBDome-Notebooks ground truth analysis repository contains notebooks 6.1.1 and 6.1.2 outlining how the XGBoost models were trained on the GT-PLIP and GT-Distance ground truth datasets. These notebooks also include details about what parameter optimisation steps were performed and tests for analysing overfitting. The GT-PLIP and GT-Distance ground truth datasets are provided on our repository as a text file (https://git.ecdf.ed.ac.uk/sgrannem/pyRBDome_Notebooks_Ground_truth_analyses/-/blob/main/analysis_results/All_combined_results.txt) and Table S5. These files contain the names of the UniProt IDs that were analysed, the PDB files we used, a list of all the amino acids and residue numbers for each protein in the PDB file, the distance of an amino acid to RNA (if available), and results from the PLIP analyses. Table S4 also contains all the prediction scores from the individual tools for each amino acid.

For the training of the XGBoost ensemble model, we normalised the scores or probabilities from each individual predictor (aaRNA, PST-PRNA, RNABindRPlus, BindUP, and DisoRDPbind) to a range between 0 and 1, where necessary. These normalised values were then used as feature values for training the models (Fig S6). In the case of FTMap data, the distances to docked molecules (in Å) were normalised to values between 0 and 1, with the highest values assigned to the shortest distances. The XGBoost model subsequently generates output files containing probabilities that indicate the likelihood of each amino acid interacting with RNA. Given that the number of RNA-interacting amino acids in the GT-PLIP and GT-Distance ground truth datasets was ~5–10%, we undersampled the majority class (i.e., non-interacting amino acids, labelled as "0's") in our training data to address the unbalanced nature of the dataset. To build the models, 80% of all structures in the ground truth datasets were used for training and 20% for testing. However, to produce the results shown in Fig 10, we trained a new XGBoost model where 50% of the data were used for training and 50% for testing to be able to do a more meaningful comparison with the RBS-ID data. Using Python's Scikit-learn and the Optuna optimisation framework (Akiba et al, 2019 Preprint), we optimised the hyperparameter for our XGBoost models. This optimisation included 10-fold cross-validation to enhance the robustness and generalisability of the models. All models, including those trained on various combinations of prediction results, are available from our repository (pyRBDome-Notebooks Ground truth analyses; 6.1 series notebooks and folder "xgboost_models").

### Analysis of predictions and cross-linking sites onto protein domains

Notebook 8.1 analyses (1) what domains were detected in cross-linked peptides and (2) which ones were enriched in the data. Notebook 9.0 extracts selected domains from the available PDB files, superimposes them, and highlights prediction scores, cross-linked peptides, and cross-linked amino acids within the superimposed structures. To be able to run notebook 9.0, we added the Linux version of MM-align (Mukherjee & Zhang, 2009) to the "programs" folder in the pyRBDome-Notebooks repository. This version was compiled on Ubuntu 22.04 and may not be compatible with later versions of Ubuntu and different operating systems. These analyses enable the user to determine whether predicted RBDs show specific cross-linking patterns, making it possible to gain information about domain RNA-binding interfaces.

## Data Availability

All the code and data analysis results are available from our GitLab repository (https://git.ecdf.ed.ac.uk/sgrannem) without restrictions. All the prediction and ground truth analysis results can be found on the repositories starting with pyRBDome-Notebooks. The pyRBDome-Core repository contains all the code required to run the pyRBDome-Notebooks Jupyter notebook files. The results of all the analyses are also available as Microsoft Excel spreadsheets in Tables S2, S3, S4, and S5.

## Supplementary Information

## Acknowledgements

We would like to thank Shaun Webb and Rasna Walia for help with RNA-BindRPlus, Songling Li and Daron Standley for help writing Python code for automatically submitting aaRNA jobs, Yun Zhou for helping with the analysis of PDB files, and Guido Sanguinetti, Andrea Weiße, and Alfredo Castello for critically reading the article. This work was supported by Medical Research Council Non-Clinical Senior Research Fellowship (MR/R008205/1 to S Granneman), Darwin Trust Award (to N Christopoulou and U Litvin), and Wellcome Trust PhD Training Fellowship awarded to H McCaughan (220406/Z/20/Z).

### Author Contributions

L-C Chu: conceptualisation, data curation, software, formal analysis, supervision, validation, investigation, visualisation, methodology, and writing—original draft.

N Christopoulou: conceptualisation, data curation, software, formal analysis, supervision, funding acquisition, validation, investigation, visualisation, methodology, and writing—original draft, review, and editing.

H McCaughan: conceptualisation, data curation, software, formal analysis, supervision, funding acquisition, validation, investigation, visualisation, methodology, and writing—original draft, review, and editing.

S Winterbourne: conceptualisation, data curation, software, formal analysis, validation, investigation, visualisation, methodology, and writing—review and editing.

D Cazzola: conceptualisation, data curation, software, formal analysis, investigation, and methodology.

S Wang: conceptualisation, data curation, software, formal analysis, validation, investigation, visualisation, and methodology.

U Litvin: conceptualisation, data curation, formal analysis, validation, investigation, visualisation, and methodology.

S Brunon: conceptualisation, data curation, formal analysis, validation, investigation, visualisation, and methodology.

PJB Harker: conceptualisation, data curation, formal analysis, investigation, visualisation, and methodology.

I McNae: conceptualisation, supervision, investigation, methodology, and project administration.

S Granneman: conceptualisation, resources, data curation, software, formal analysis, supervision, funding acquisition, validation, investigation, visualisation, methodology, project administration, machine-learning tool development, and writing—original draft, review, and editing.

## Conflict of Interest Statement

The authors declare that they have no conflict of interest.

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
