## [Reviewer comments · Life Science Alliance]

Life Science Alliance

pyRBDome: A comprehensive computational platform for enhancing RNA-binding proteome data

Liang-Cui Chu, Niki Christopoulou, Hugh McCaughan, Sophie Winterbourne, Davide Cazzola, Shichao Wang, Ulad Litvin, Salomé Brunon, Patrick Harker, Iain McNae, and Sander Granneman

DOI: <https://doi.org/10.26508/lsa.202402787>

Corresponding author(s): Sander Granneman, University of Edinburgh and Sander Granneman, University of Edinburgh

Review Timeline:

Submission Date:	2024-04-22
Editorial Decision:	2024-04-23
Revision Received:	2024-06-24
Editorial Decision:	2024-07-03
Revision Received:	2024-07-15
Accepted:	2024-07-16

Transaction Report:

Please note that the manuscript was previously reviewed at another journal and the reports were taken into account in the decision-making process at *Life Science Alliance*.

Reviewer #1

Report for Author:

Summary:

In this manuscript, the authors develop a computational platform named pyRBDome to predict RNA-binding sites (RBSs) in proteins of interest. On one hand, the authors aggregate multiple RBS-predicting software (i.e. RNABindRPlus, FTMap) to build a new one with enhanced performance. On the other side, the authors enable the integration of mass spectrometric datasets mapping RBSs (i.e. RBS-ID in Bae et al., 2020) or RBDs (RBDmap in Castello et al., 2016) proteome-wide. Both analyses are compared against a structural data-based ground-truth datasets (GT-Distance, GT-PLIP).

General remarks:

Overall, the authors present a useful computational tool to predict the RNA-binding properties of a desired protein. Given the recent advances in the field of protein structure prediction (i.e. Alphafold DB, Colabfold, RoseTTAFold), pyRBDome will serve as a great starting point for those who move on to analyze RNA-protein interactions. In addition, the implementation in python allows easy interpretation and adjustments by the users.

Nevertheless, its novelty and the usefulness in the identification of bona-fide RBD/RBS seem to be in question. While the authors show that RBS-ID results contain certain biases and mis-identified many of the RBD, based on the assumption that structure information of protein-RNA complex can serve as "ground truth", they did not show to what extent, if possible, pyRBDome can significantly improve or at least compensate in the same situation. It would greatly improve the manuscript if the authors demonstrate how that can be done. In theory, the tool would be useful in the situations when largely unknown organisms or unconventional RBPs are the focus of study. Authors does not seem to provide such evidences and seemed to have largely focused on the validating the tool's function with well-known RBPs in human.

In sum, while the usefulness and applicability/convenience of the tool is in no doubt, the study largely lacks new information. I would recommend the authors to demonstrate how it can also function in the identification of bona-fide RBD/RBS in both novel organism and RBD. In addition, it would be important to show that, beside the potential benefit that has already been outlined, pyRBDome can be a significant alternative and/or compensate RBS-ID.

Major points:

1. The manuscript at the current state is a bit confusing to understand. Some re-organization of the figures and text would help better understand the utility of pyRBDome.

Firstly, a head-start schematic figure on the pyRBDome platform would help the audience get a quick overview on its utility. For instance, the authors can place a flow chart describing input types, processing nodes, and output types at the beginning of the manuscript.

Secondly, the manuscript would be better taken up if the sections on [1] Ground truth generation & demonstration of pyRBDome output (Figs. 1, 2 & 4) [2] RBS-predicting software (Figs. 3 & 8) and [3] integration of RBS-mapping databases (Figs. 5-7) are explicitly divided. For this, the authors can re-arrange the figures and the corresponding text accordingly.

2. Page 12 line 315: Authors trained the algorithm based on the structural information but there are many RBPs without such protein-RNA complex structures. While BindUP, RNABindRPlus, and aaRNA function best for such RBPs, it may not be true for the unconventional RBPs such as metabolic enzymes.

3. Page 17 line 412-415: Author's claim seems to be an overstatement. The experiments were done in the live cells, in which there are dynamic interactions between RNA and protein. Both RNA and protein may exist in distinct form or state, potentially in complex to different molecules or modified. The reaction may also have been relatively unstable/transient, thus not shown in the structure of the molecules.

4. Can the authors illustrate the use pyRBDome on a deep RBS-profiling dataset on a single target protein? (i.e. the spCas9 data in Bae et al., 2020)

5. The authors can also compare the results from integrating peptide-level datasets (i.e., RBDmap, RBR-ID) with amino acid-level datasets (i.e., RBS-ID). This can be done with an example protein/domain.

Minor points:

1. As in Fig. 2A, the authors can add a structure that highlights RBSs and/or peptides (in reds and yellows?) identified by mass spectrometric databases, if available, as a part of pyRBDome output.

2. The legend Fig. 8 is overwritten with that of Fig. 7.
3. In some parts of the text, RBS-ID is misspelled as RBD-ID. Please double check the names of previous resources.

Reviewer #2

Report for Author:

The authors Chu et al. present a comprehensive software platform to study RNA-binding proteome data.

I understand that the implementation consists of an enormous number of Jupyter notebooks and depends on external tools and web services.

The basic input requirement is a list of protein IDs, which are analyzed for RNA binding sites. Additional sequence information based on experimental assay technology can be provided as well to pinpoint candidates more specifically.

The overall prediction performance of single existing tools is improved through an ensemble learning approach, which combines sequence- and structure-based prediction tools.

Generally, the performance heavily depends on the training set (AUPRC: 0.61 vs 0.41, Figure 3)

The authors then show that their software could be used on less annotated organisms such as *S. aureus* to derive some insights on RNA interacting amino acids.

Another finding is that information from protein structures does not necessarily agree with information from cross-linking experiments

Taken together, pyRBDome is a toolkit to generate hypotheses on the precise position of RNA-binding residues from RNA-binding proteome experiments.

Honestly, it was a little difficult for me to manage my expectations with regards to the manuscript content.

Naively, one could use the software as a tool for RBP prediction. A more reasonable use case would be the annotation of candidate RBPs from experiments or a somewhat pre-defined list by an expert RNA biologist.

However, it is unclear if and how the method performs on different known RNA-binding domains / folds. There are no detailed results given on this aspect.

IMHO, the software would predict RNA-interacting residues on any given protein list, is hard to execute and requires expert / domain knowledge to make sense of the final output.

Reviewer #3 Review

Report for Author:

In this study, Chu et al describe the development of pyRBDome, a computational pipeline for interpreting RNA-binding proteome data. It aligns experimental results with RNA-binding site predictions and high-resolution structural data to identify genuine RNA binders in experimental datasets. The pipeline also enhances the sensitivity and specificity of RNA-binding site detection through the training of new ensemble machine-learning models. The authors highlight the limitations of structural data as benchmarks and position pyRBDome as a valuable alternative for increasing confidence in RNA-binding proteome datasets. The manuscript is well-written and easy to follow and most of the results are convincing. I find this work to be valuable for the field and have only a few suggestions to improve the manuscript:

Comments:

1. Can the authors add a feature that will allow pyRBDome to predict the RNA sequences bound by the RNA-binding proteins? If this is beyond the scope of this manuscript, the authors can at least address this option in the discussion.

2. The authors use many individual tools as part of or in addition to the pyRBDome throughout the manuscript. I found it sometimes hard to follow all the acronyms. I would suggest adding a table that summarizes the main tools used and/or developed in this study.
3. Line 432: "Figures 7" should be "Figure 7"?
4. Line 465" Why is it written "(Model)"?
5. Lines 559-571: Can the authors comment on what is expected to happen to the UV cross-linking efficiency if one of the amino acids found in indirect proximity is mutated?
6. Can the authors discuss what is the smallest protein pyRBDome will be able to analyze?

April 23, 2024

Re: Life Science Alliance manuscript #LSA-2024-02787-T

Dr Sander Granneman
University of Edinburgh
Centre for Synthetic and Systems Biology (SynthSys)
Max Born Crescent
CH Waddington Building, room 3.06
Edinburgh, MidLothian EH9 3BF
United Kingdom

Dear Dr. Granneman,

Thank you for submitting your manuscript entitled "pyRBDome: A comprehensive computational platform for enhancing and interpreting RNA-binding proteome data" to Life Science Alliance. We invite you to submit a revised manuscript as indicated in your draft rebuttal.

Thank you for this interesting contribution to Life Science Alliance. We are looking forward to receiving your revised manuscript.

Sincerely,

Eric Sawey, PhD
Executive Editor
Life Science Alliance
<http://www.lsa-journal.org>

B. MANUSCRIPT ORGANIZATION AND FORMATTING:

We would like to thank the reviewers for their valuable and constructive feedback. In response to the reviewers' comments, we have made a substantial number of changes to the manuscript. Firstly, we have included results using the latest pyRBDome model (version 0.24), which now incorporates predictions from PST-PRNA. These predictions were already included in our Git repository, and PST-PRNA was discussed as a future implementation for pyRBDome in the previous version of the manuscript. However, we have now included these new analyses to ensure that our paper discusses the latest pyRBDome developments.

As requested by the reviewers, we have also reorganised the text to improve its flow. We added a new Figure 1 outlining the key steps of the pipeline and the output it generates. Additionally, as requested, we now use the spCas9 RBS-ID data to showcase the outputs of the pyRBDome pipeline.

Furthermore, we have included new data (Fig. 10) showing that pyRBDome can be a viable alternative to RBD-ID by demonstrating that it outperforms this experimental approach on all analysed metrics.

Reviewer #1:

Summary:

In this manuscript, the authors develop a computational platform named pyRBDome to predict RNA-binding sites (RBSs) in proteins of interest. On one hand, the authors aggregate multiple RBS-predicting software (i.e. RNABindRPlus, FTMap) to build a new one with enhanced performance. On the other side, the authors enable the integration of mass spectrometric datasets mapping RBSs (i.e. RBS-ID in Bae et al., 2020) or RBDs (RBDmap in Castello et al., 2016) proteome-wide. Both analyses are compared against a structural data-based ground-truth datasets (GT-Distance, GT-PLIP).

General

remarks:

Overall, the authors present a useful computational tool to predict the RNA-binding properties of a desired protein. Given the recent advances in the field of protein structure prediction (i.e. AlphaFold DB, Colabfold, RoseTTAFold), pyRBDome will serve as a great starting point for those who move on to analyze RNA-protein interactions. In addition, the implementation in python allows easy interpretation and adjustments by the users.

1. Nevertheless, its novelty and the usefulness in the identification of bona-fide RBD/RBS seem to be in question. While the authors show that RBS-ID results contain certain biases and mis-identified many of the RBD, based on the assumption that structure information of protein-RNA complex can serve as "ground truth", they did not show to what extent, if possible, pyRBDome can significantly improve or at least compensate in the same situation. It would greatly improve the manuscript if the authors demonstrate how that can be done. In theory, the tool would be useful in the situations when largely unknown organisms or unconventional RBPs are the focus of study. Authors does not seem to provide such evidences and seemed to have largely focused on the validating the tool's function with well-known RBPs in human.

Response: We are grateful for the constructive review of our manuscript and for acknowledging the utility of pyRBDome as a computational tool for predicting RNA-binding properties of proteins. With respect to the reviewers' point about how pyRBDome can compensate for biases in RBS-ID data or act as a viable alternative, we agree that we did not discuss this with sufficient detail in the manuscript. While the pyRBDome pipeline is unable to 'compensate' for the biases associated with RBS-ID, based on our studies, we do make several recommendations when it comes to the interpretation of the RBS-ID data. Given our observation that analysing cross-linked amino acids may not be the most reliable source for identifying RBS, and the poor overlap with the crystal structure data, we recommend that users

focus on characterising amino acids in cross-linked peptides that have high XGBoost-predicted RNA-binding propensities, particularly those that are positively charged or aromatic. This is now discussed on page 17 of the revised manuscript. Can pyRBDome be a viable alternative for RBS-ID? We would say yes: as detailed below, in the new Figure 10 we show that the pyRBDome performs better at identifying RBS compared to the cross-linked amino acids reported by the RBS-ID dataset. These new analyses imply that pyRBDome is not only a good complementary approach, but also a good alternative for RBS-ID, should a lab not be equipped to perform such experiments. Notably, we also make clear on page 16 and 17 that one of the limitations of the current study is that the model is trained on proteins that mostly have well-characterised RNA-binding domains and it is unclear whether the pipeline will perform better on less unusual or less-studied RNA-binding domains. This is largely because we cannot perform a statistical evaluation of pyRBDome and RBS-ID performance on unusual/uncharacterised RBPs because of the scarcity of mechanistic insights into how these domains bind RNA (see page 17). For these proteins it would be valuable to have *both* RBS-ID and pyRBDome data.

Regarding the concerns raised about the novelty and usefulness of pyRBDome in identifying *bona-fide* RBD/RBS, we acknowledge that our focus has primarily been on validating the tool's function with well-known human and *S. aureus* RBPs. This decision was deliberate as it allowed us to conduct a statistical comparison of pyRBDome results with known experimental data, thereby providing a meaningful assessment of the tool's performance. We agree that seeing how pyRBDome could perform in identifying RBPs from largely unknown organisms or unconventional RBPs would be of interest. It will generate predictions for sure. This is also exactly why we chose to apply pyRBDome to our *S. aureus* RBPome dataset (Chu et al Nature Communications 2022), because it contained many proteins with domains not previously found associated with RNA. Many of these putative RBPs also did not have homologues in other bacteria. However, as outlined above, it will not be possible to determine how accurate many of these predictions are and therefore to what extent they are meaningful. In situations where tools are also used to make predictions on a large scale, experimental validation (for example using CLIP) remains essential. However, this is not always at a feasible at the scale needed. Thus, while we reported pyRBDome prediction results for many unusual RBPs identified in *S. aureus*, we cannot provide a statistical metric for the performance of pyRBDome in these scenarios. We now also make this explicitly clear on page 17 of the revised manuscript. Even so, we believe that the tool's predictions can still serve to generate valuable hypotheses for further experimental investigation. An example is illustrated in Rebuttal Figure 1 where we used pyRBDome to predict an RBS for an unusual RBP and perform some validation analyses. These results are discussed below.

2. In sum, while the usefulness and applicability/convenience of the tool is in no doubt, the study largely lacks new information. I would recommend the authors to demonstrate how it can also function in the identification of bona-fide RBD/RBS in both novel organism and RBD. In addition, it would be important to show that, beside the potential benefit that has already been outlined, pyRBDome can be a significant alternative and/or compensate RBS-ID.

Response: We thank the reviewer for the suggestions. With respect to the reviewer's recommendation, yes, we would argue that pyRBDome is very useful in identifying putative RNA-binding sites in novel or unusual domains, even though it has been trained on mostly data from canonical RBPs. To illustrate this point, below we show some data from a different project that is ongoing in the lab. Niki Christopoulou (one of the authors of this manuscript) discovered that proteins involved in peptidoglycan synthesis were highly enriched in our *Staphylococcus aureus* RNA-binding proteome data (Chu et al., Nature Communications 2022). This includes the PBP2a peptidoglycan synthesis enzyme that confers resistance to β -lactam antibiotics in methicillin-resistant *S. aureus* (MRSA). She subsequently verified RNA-binding *in vivo* using the CLIP approach and identified numerous RNA-binding ligands.

However, it was initially unclear which region of the protein interacted with the RNA, as PBP proteins have never been previously associated with RNA binding. Therefore, we utilised the pyRBDome pipeline to predict putative RNA-binding sites in PBP2a and other PBPs identified in our *S. aureus* RBPome datasets. The PBP2a prediction results are presented in Rebuttal Figure 1 below. To our surprise, these predictions suggested that the active site region of this enzyme (transpeptidase domain; highlighted in the structure) might also be involved in RNA binding. To substantiate the pyRBDome prediction results, we employed a docking tool named H-dock (Yan et al., NAR, 2017) to model short RNA sequences (UCUU) on the surface of PBP2a. This revealed that a significant proportion of the RNA molecules were docked in regions predicted to bind RNA by pyRBDome. Additionally, Niki conducted CLIP experiments to identify PBP2a RNA-binding sequences *in vivo*. This experiment identified the non-coding RNA Teg104 as a potential RNA substrate. We subsequently asked AlphaFold3 (Abramson et al., Nature, 2024) to dock the Teg104 sequence on the PBP2a structure. Strikingly, these data were also consistent with the pyRBDome predictions, with Teg104 docked on the same surface patch predicted to bind RNA.

Using the pyRBDome prediction results shown in Rebuttal Figure 1, we then generated several PBP2a amino acid substitutions that we predicted would block RNA-binding. The results from these analyses are shown in Rebuttal Figure 2. This enabled us to identify PBP2a amino acids (K436 and R445) that are important for RNA-binding (Rebuttal Figure 2B and 2C). Note that we also generated mutations in regions *not* predicted to bind RNA. These mutants did not show significant reductions in RNA interactions *in vivo*.

In our opinion, these data convincingly demonstrate that pyRBDome can credibly predict RNA-binding regions, even in proteins never found to be associated with RNA before.

However, the data described here, constituting over three years of hard work, are destined to be published in a separate manuscript describing the RNA-binding properties of PBP2a. But we hope these data address the reviewers' concern.

While these data showcase the potential of pyRBDome, to truly assess the “performance” of pyRBDome on *unusual* RNA-binding domains, many individual proteins would need to be analysed in a similar way. This would take years to complete and would be an entirely new project on its own. We feel that this is therefore beyond the scope of the current manuscript.

[Figure removed by editorial staff per authors' request]

[Figure removed by editorial staff per authors' request]

Regarding demonstrating whether pyRBDome is a significant alternative to RBS-ID, we believe that this is the case. The issue with RBS-ID and related UV cross-linking methods lies in the stochastic nature of UV cross-linking, which samples only a fraction of all RNA-binding residues. Consequently, this leads to many false negatives in the data, as it fails to detect numerous true RNA-binding residues. Comparing the cross-linking data and the predictions from our XGBoost model with the ground truth datasets vividly illustrates this point (see Rebuttal Figure 3 below, now Figure 10 in the main manuscript).

We performed these analyses on both the spCas9 RBS-ID data that was published alongside the RBS-ID data, as well as on proteins in the RBS-ID dataset for which we had ground truth data. These results demonstrate that when using both our GT-PLIP and GT-Distance ground truth datasets, XGBoost significantly outperforms RBS-ID across all tested performance metrics, both on the spCas9 data and all the human proteins analysed. A description of the performance metrics used in this figure is provided in the figure legend. Notably, RBS-ID exhibits poor precision and recall, detecting only between 1-13% of the RNA-binding residues, depending on the ground truth dataset used. Furthermore, precision in the detection of RNA-binding amino acids is also considerably poorer when compared to the XGBoost model. In fact, the Matthews Correlation Coefficient (MCC) for the RBS-ID compared with ground truth datasets shows values near zero, suggesting that the RBS-ID data perform similarly to randomly picked amino acids. Importantly, however, these results are not unexpected: given the stochastic nature of UV cross-linking, one would expect to see poor recall as only a small number of the actual RNA-binding sites are detected. The low precision is a concern, as only 12% of the spCas9 cross-links mapped to amino acids binding RNA in the crystal structure (Rebuttal Figure 3A), and about one-third of the reported cross-links were in regions that are within hydrogen-bonding distance of RNA in the structure (Rebuttal Figure 3B). However, as outlined in the paper, we question whether the structural data is really useful to properly benchmark RBS-ID. These results are now included in the new Figure 10 and discussed on pages 16 and 17 of the revised manuscript.

A GT-PLIP

B GT-Distance

Rebuttal Figure 3. Performance metrics used when comparing the RBS-ID cross-linked amino acids with the XGBoost predictions and ground truth datasets. Colours closer to red indicate high performance, whereas colours closer to blue indicate poor performance. **A.** Comparing the XGBoost predictions to the spCas9 and proteome RBS-ID cross-linking data using GT-PLIP data generated from published protein-RNA structures. **(B.** Same as in A) but now considering amino acids that are within 4.2Å from RNA as RNA-binding residues. Explanation of the performance metrics: **Accuracy:** the frequency of model predictions that are correct. Note that mostly the non-RNA binding amino acid residues are correctly predicted. **Precision:** measures what fraction of the items predicted as positive by RBS-ID and XGBoost are also positive in the ground truth datasets. **Recall:** measures what fraction of the positive items in the ground truth datasets were identified correctly by RBS-ID and our XGBoost model. **F1 Score:** the harmonic mean of precision and recall. F1 score reaches its best value at 1 (perfect precision and recall) and worst at 0. **MCC:** the Matthews Correlation Coefficient considers true and false positives and negatives and returns a value between -1 and +1, where +1 indicates perfect prediction, 0 indicates random prediction, and -1 indicates total disagreement between prediction and observation.

Major points:

3. The manuscript at the current state is a bit confusing to understand. Some re-organization of the figures and text would help better understand the utility of pyRBDome. Firstly, a head-start schematic figure on the pyRBDome platform would help the audience get a quick overview on its utility. For instance, the authors can place a flow chart describing input types, processing nodes, and output types at the beginning of the manuscript. Secondly, the manuscript would be better taken up if the sections on [1] Ground truth generation & demonstration of pyRBDome output (Figs. 1, 2 & 4) [2] RBS-predicting software (Figs. 3 & 8) and [3] integration of RBS-mapping databases (Figs. 5-7) are explicitly divided. For this, the authors can re-arrange the figures and the corresponding text accordingly.

Response: Agreed, and we have made the suggested changes to improve the flow. We now include a new figure showing a brief overview of the pipeline in Figure 1. Based on the reviewers' recommendations, we have shuffled around some of the sections with the aim of improving the flow.

4. Page 12 line 315: Authors trained the algorithm based on the structural information but there are many RBPs without such protein-RNA complex structures. While BindUP, RNABindRPlus, and aaRNA function best for such RBPs, it may not be true for the unconventional RBPs such as metabolic enzymes.

Response: While we acknowledge that for most RNA-binding metabolic enzymes, structures with RNA are not readily available, it's important to note that such structures are not critical for training purposes nor needed for predicting RNA-binding residues within metabolic enzymes. These predictions leverage a wide array of both sequence and structural features and can accurately identify RNA-binding residues without prior knowledge of known RNA-binding sites, including those in metabolic enzymes.

While some tools do rely on structural information for training purposes, others, such as RNABindRPlus and DisoRDPbind employed by pyRBDome, solely utilise sequence information for their predictions. This includes factors such as the electrostatic properties of amino acid residues, conservation of residues, and, in some cases, even the properties of neighbouring amino acids.

Regarding BindUP, the assertion that it is mostly suited for RBPs with available protein-RNA structures is incorrect. While BindUP does require structural information for the protein (but not the RNA), it does not search for specific folds but rather identifies patches of positively charged residues on the protein surface to make predictions. Notably, BindUP also does not rely on homology between protein sequences for its predictions. Therefore, we included this tool in our analyses because it is better equipped to predict RNA-binding regions in proteins with non-classical RNA-binding domains, such as metabolic enzymes. For example, in the case of our PBP2a data, one of the main reasons why we focused on the active site was because BindUP (as well as RNABindRPlus) detected a positively charged amino acids specifically in this region predicted to bind RNA. These results provided the basis of our PBP2a mutational analyses.

5. Page 17 line 412-415: Author's claim seems to be an overstatement. The experiments were done in the live cells, in which there are dynamic interactions between RNA and protein. Both RNA and protein may exist in distinct form or state, potentially in complex to different molecules or modified. The reaction may also have been relatively unstable/transient, thus not shown in the structure of the molecules.

Response: We agree, and this is also what we were trying to bring across in our manuscript. However, upon rereading the relevant text, it was clear that this aspect could have been articulated more effectively. Therefore, we propose the following amendment:

“These data therefore reinforce the idea that, when comparing the experimental data to existing structural data, UV cross-linking does not always capture amino acids directly binding to RNA, but that they are generally closer to RNA molecules.”

To:

“These data therefore reinforce the idea that UV cross-linking does not always capture amino acids that in static protein-RNA structures directly bind RNA, but that they are generally closer to these RNA molecules in these structures

This is now mentioned on pages 14 and 15 of the revised manuscript.

6. Can the authors illustrate the use pyRBDome on a deep RBS-profiling dataset on a single target protein? (i.e. the spCas9 data in Bae et al., 2020)

Response: Agreed. We now use the spCas9 data as examples for illustrating the use of pyRBDome. These new results are shown in the new Figures 2, 3 and 10 and Figure EV3.

7. The authors can also compare the results from integrating peptide-level datasets (i.e., RBDmap, RBR-ID) with amino acid-level datasets (i.e., RBS-ID). This can be done with an example protein/domain.

Response: Using spCas9 as an example, we now illustrate in the revised Figure 3B that pyRBDome can also be used to show peptide-level data.

Minor points:

8. As in Fig. 2A, the authors can add a structure that highlights RBSs and/or peptides (in reds and yellows?) identified by mass spectrometric databases, if available, as a part of pyRBDome output.

Response: Agreed. We have now made a new Figure 3 showing the cross-linking data for spCas9, where we also highlight cross-linked peptides we generated and the cross-linked amino acids. We chose spCas9 as an example due to the availability of extensive RBS-ID cross-linking data and because structural data of this protein in complex with RNA is also available. We believe this example nicely demonstrates how pyRBDome can integrate various data sources to identify and visualise RNA-binding sites and peptides.

In the new figure, cross-linked peptides are highlighted in red as spheres (Fig. 3C), and the peptides are highlighted in green (Fig. 3B).

9. The legend Fig. 8 is overwritten with that of Fig. 7.

Response: Corrected.

10. In some parts of the text, RBS-ID is misspelled as RBD-ID. Please double check the names of previous resources.

Response: Corrected.

Reviewer #2:

The authors Chu et al. present a comprehensive software platform to study RNA-binding proteome data.

I understand that the implementation consists of an enormous number of Jupyter notebooks and depends on external tools and web services.

The basic input requirement is a list of protein IDs, which are analyzed for RNA binding sites. Additional sequence information based on experimental assay technology can be provided as well to pinpoint candidates more specifically.

1. The overall prediction performance of single existing tools is improved through an ensemble learning approach, which combines sequence- and structure-based prediction tools.

Response: This statement is inaccurate. The performance of the individual tools does not improve. Instead, the performance of our new machine learning model significantly surpasses that of existing tools when the machine learning model is trained on the combined predictions from those existing tools. The XGBoost model not only learns based on the successful predictions made by the individual tools, but also learns from their mistakes. This distinction is crucial to understanding the core contribution of our manuscript.

2. Generally, the performance heavily depends on the training set (AUPRC: 0.61 vs 0.41, Figure 3)

Response: Indeed, we used two different training datasets for a specific reason. The PLIP ground truth datasets exclusively contain amino acids that directly bind RNA based on available structures. However, for developing models, this dataset presented two challenges: firstly, the number of proteins included was relatively low, and secondly, the proportion of amino acids binding RNA in the analysed proteins was less than 5%. Such an "unbalanced" dataset can pose challenges in building robust machine learning models.

To address these challenges, we generated a second ground truth dataset containing a larger number of proteins. This dataset includes amino acids within a 4.2Å distance of RNA, a common threshold used in building training datasets for predicting RNA-binding residues in proteins. While many groups typically use a cut-off of 5Å, we opted for amino acids within hydrogen-bonding distance (4.2Å) for improved accuracy and specificity. This distinction underscores the importance of selecting appropriate training data to develop accurate and reliable machine learning models for predicting RNA-binding residues in proteins.

Note that in the revised manuscript we managed to further increase the average precision to 0.7 by including data from the PST-PRNA prediction algorithm. These results are now shown in the revised Figure 6.

3. The authors then show that their software could be used on less annotated organisms such as *S. aureus* to derive some insights on RNA interacting amino acids. Another finding is that information from protein structures does not necessarily agree with information from cross-linking experiments

Response: We would also like to emphasise that our analyses represent the most comprehensive investigations to date. We are the first to systematically examine the overlap between RBS-ID datasets and known structural data. Previous studies have typically focused on a limited number of proteins, whereas our analysis spans hundreds, enabling us to draw conclusions supported by robust statistical analyses. This distinction underscores the breadth

and depth of our research, providing valuable insights into the relationship between RBS-ID datasets and known structural information in RNA-binding proteins.

Taken together, pyRBDome is a toolkit to generate hypotheses on the precise position of RNA-binding residues from RNA-binding proteome experiments.

Honestly, it was a little difficult for me to manage my expectations with regards to the manuscript content.

4. Naively, one could use the software as a tool for RBP prediction. A more reasonable use case would be the annotation of candidate RBPs from experiments or a somewhat pre-defined list by an expert RNA biologist.

Response: This statement aligns perfectly with the intended purpose of pyRBDome. Users have the flexibility to utilise it either to bolster their RBS-ID/RBDMap dataset or to study proteins of interest, including novel ones. This versatility is highlighted in our introduction, where we outline the various applications and benefits of pyRBDome. But the user does not necessarily need to be an expert RNA biologist.

5. However, It is unclear if and how the method performs on different known RNA-binding domains / folds. There are no detailed results given on this aspect.

Response: It is unclear precisely what the reviewer is requesting here. The data have been trained on a very large number of different proteins, protein domains and folds, and the model makes predictions even for non-canonical RNA-binding domains. Additionally, to gain insights into how well a model performs on individual domains or folds, we require many existing structures of these domains in complex with RNA. Unfortunately, for many individual domains, such data is not readily available or not available in the quantities needed to perform a meaningful statistical analysis. Moreover, it is also not straightforward to perform a per domain comparison between the experimental and pyRBDome data: the RBS-ID and RBDMap data that we analysed mostly return cross-links in RRM domains, which enabled us to perform a statistical analysis comparing the experimental data with the predictions (Figure 5). However, for all the other domains identified in the data, the number of cross-links was too low to make meaningful comparisons (see KH domain example in Figure EV6).

6. IMHO, the software would predict RNA-interacting residues on any given protein list, is hard to execute and requires expert / domain knowledge to make sense of the final output.

Response: While we acknowledge that some domain knowledge is necessary to run our analyses, this holds true for all published bioinformatics methods that we have reviewed. Any researcher with an interest in studying protein-RNA interactions and some experience in running Python code will possess sufficient domain knowledge to utilise pyRBDome. This includes undergraduate and MSc students who have contributed to the development of the pipeline.

We disagree with the assertion that it will be challenging to interpret the outputs produced by pyRBDome. On the contrary, pyRBDome was designed to facilitate easier interpretation of results by presenting predictions in both 3D and the protein sequence format. Even for those who may prefer not to rely solely on the predictions of our XGBoost machine learning model, one can easily identify residues that are most consistently predicted to bind RNA by multiple algorithms and proceed with mutational studies in their preferred RNA-binding protein.

Reviewer #3:

In this study, Chu et al describe the development of pyRBDome, a computational pipeline for interpreting RNA-binding proteome data. It aligns experimental results with RNA-binding site predictions and high-resolution structural data to identify genuine RNA binders in experimental datasets. The pipeline also enhances the sensitivity and specificity of RNA-binding site detection through the training of new ensemble machine-learning models. The authors highlight the limitations of structural data as benchmarks and position pyRBDome as a valuable alternative for increasing confidence in RNA-binding proteome datasets. The manuscript is well-written and easy to follow and most of the results are convincing. I find this work to be valuable for the field and have only a few suggestions to improve the manuscript:

Comments:

1. Can the authors add a feature that will allow pyRBDome to predict the RNA sequences bound by the RNA-binding proteins? If this is beyond the scope of this manuscript, the authors can at least address this option in the discussion.

Response: This is a great idea, but it presents significant challenges in implementation. However, there are methods available that predict RNA-binding motifs based on amino acid sequences. Incorporating this into pyRBDome would involve a substantial amount of additional work to test the many different algorithms that are available and analyses of new dataset types. While this extends beyond the scope of the current manuscript, we appreciate the suggestion and now discuss it in the revised version of the manuscript on page 21.

2. The authors use many individual tools as part of or in addition to the pyRBDome throughout the manuscript. I found it sometimes hard to follow all the acronyms. I would suggest adding a table that summarizes the main tools used and/or developed in this study.

Response: This is a great suggestion. This information is now provided in the new Table 1 in the revised manuscript.

3. Line 432: "Figures 7" should be "Figure 7"?

Response: Corrected.

4. Line 465" Why is it written "(Model)"?

Response: We wanted to indicate that also AlphaFold2 structures were downloaded for those proteins where no structural data was available. We have now changed the phrasing on page 13 to make this clearer:

"Structures (from rcsb.org or AlphaFold2) for the top 200 enriched proteins were analysed by the pipeline and the results are available on our GitLab repository."

5. Lines 559-571: Can the authors comment on what is expected to happen to the UV cross-linking efficiency if one of the amino acids found in indirect proximity is mutated?

Response: This is an interesting idea. Given that the indirect cross-linking is likely stochastic, the cross-link would presumably move to another neighbouring amino acid if the chemical environment is correct. The Knörlein paper published in Nature Communications (2022) clearly shows that the photo reactivity of the nucleotide is very important so simply making substitutions may not provide a clear answer.

6. Can the authors discuss what is the smallest protein pyRBDome will be able to analyze?

Response: There is no sequence length limit for the pipeline.

July 3, 2024

RE: Life Science Alliance Manuscript #LSA-2024-02787-TR

Prof. Sander Granneman
University of Edinburgh
Centre for Engineering Biology
Max Born Crescent
CH Waddington Building, room 3.06
Edinburgh, MidLothian EH9 3BF
United Kingdom

Dear Dr. Granneman,

Thank you for submitting your revised manuscript entitled "pyRBDome: A comprehensive computational platform for enhancing RNA-binding proteome data". We would be happy to publish your paper in Life Science Alliance pending final revisions necessary to meet our formatting guidelines.

- please be sure that the authorship listing and order is correct
- please upload all figure files as individual ones, including the supplementary figure files; all figure legends should only appear in the main manuscript file
- LSA allows supplementary figures, but not EV Figures; please update your callouts for the Supplementary Figures in the manuscript Fig EV1A=Fig S1A; while supplementary figures use the system supplementary Fig S1
- please incorporate the supplementary references into the main reference list
- please add your main, supplementary figure, and table legends to the main manuscript text after the references section
- please add callouts for Figures S5A-B and S7A-B to your main manuscript text

LSA now encourages authors to provide a 30-60 second video where the study is briefly explained. We will use these videos on social media to promote the published paper and the presenting author (for examples, see <https://docs.google.com/document/d/1-UWCfbE4pGcDdcgzcmiuJI2XMBJnxKYeqRvLLrLS08s/edit?usp=sharing>). Corresponding or first-authors are welcome to submit the video. Please submit only one video per manuscript. The video can be emailed to contact@life-science-alliance.org

A. FINAL FILES:

B. MANUSCRIPT ORGANIZATION AND FORMATTING:

Sincerely,

July 16, 2024

RE: Life Science Alliance Manuscript #LSA-2024-02787-TRR

Prof. Sander Granneman
University of Edinburgh
Centre for Engineering Biology
Max Born Crescent
CH Waddington Building, room 3.06
Edinburgh, MidLothian EH9 3BF
United Kingdom

Dear Dr. Granneman,

Thank you for submitting your Methods entitled "pyRBDome: A comprehensive computational platform for enhancing RNA-binding proteome data". It is a pleasure to let you know that your manuscript is now accepted for publication in Life Science Alliance. Congratulations on this interesting work.

DISTRIBUTION OF MATERIALS:

Again, congratulations on a very nice paper. I hope you found the review process to be constructive and are pleased with how the manuscript was handled editorially. We look forward to future exciting submissions from your lab.

Sincerely,
